# You Are What You Train: Rethinking Training Data Quality, Targets, and Architectures for Universal Speech Enhancement

## Abstract

Universal Speech Enhancement (USE) aims to restore the quality of diverse degraded speech while preserving fidelity. Despite recent progress, several challenges remain. In this paper, we address three key issues. (1) In speech dereverberation, the conventional use of early-reflected speech as the training target simplifies model training, but we found that it still harms perceptual quality. We therefore apply time-shifted anechoic clean speech as a simple yet more effective target. (2) Regression models preserve fidelity but produce over-smoothed outputs under severe degradation, while generative models improve perceptual quality but risk hallucination. We introduce a two-stage framework that effectively combines the strengths of both approaches, inspired by a recent theoretical finding. (3) We study the trade-off between training data scale and quality, a critical factor when scaling to large, imperfect corpora. Experimental results demonstrate that using time-shifted anechoic clean speech as the learning target significantly improves both speech quality and downstream automatic speech recognition (ASR) performance, while the two-stage framework further boosts quality without compromising fidelity. In addition, our model demonstrates strong language-agnostic capability, making it well-suited for enhancing training data in other speech generative tasks. To ensure reproducibility, the code will be made publicly available upon acceptance of the paper. Several enhanced real noisy speech examples are provided on the demo page: `https://anonymous.4open.science/w/USE-5232/`

## 1 INTRODUCTION

In universal speech enhancement (USE) (Serrà et al., 2022), the goal is to improve the intelligibility and perceptual quality of diverse degraded speech while ensuring that all other factors (e.g., speaker identity, emotion, accent) remain unchanged (Babaev et al., 2024). Recent efforts have increasingly shifted from task-specific approaches toward models that generalize across domains and conditions. For instance, VoiceFixer (Liu et al., 2021) adopts a ResUNet-based analysis stage coupled with a neural vocoder-based synthesis stage. Similarly, MaskSR (Li et al., 2024) employs a masked generative modeling objective to handle comparable conditions. More recently, AnyEnhance (Zhang et al., 2025a) introduced prompt-guidance and a self-critic mechanism, offering a unified framework capable of mitigating diverse degradations across both speech and singing voices. To advance the generalization of USE, the URGENT 2025 Challenge was introduced at Interspeech 2025 (Saijo et al., 2025). The Challenge provides training data from diverse sources with varying quality and covers seven types of distortions (additive noise, reverberation, clipping, bandwidth limitation, codec artifacts, packet loss, and wind noise), spanning multiple sampling rates (8, 16, 22.05, 24, 32, 44.1, and 48 kHz) across five languages (English, German, French, Spanish, and Chinese). This paper follows the Challenge setup and investigates three critical aspects of our proposed USE pipeline: (1) training targets, (2) model architecture, and (3) the role of training data quality.

**Training targets**: Six out of seven speech signal distortions considered in this Challenge use the original anechoic clean speech as the learning target. However, the training target for **reverberation** is the early-reflected speech signal, obtained by convolving the anechoic clean speech with the early reflection component of the room impulse response (RIR). This choice of the learning target follows a common convention, as many studies have shown that early reflections are particularly difficult to

remove, and attempting to do so often introduces excessive artifacts into the enhanced speech (Valin et al., 2022; Zhou et al., 2023; Zhao et al., 2020). Nevertheless, we found that early reflections still degrade both perceptual quality and machine intelligibility (for downstream ASR). In addition, early reflections present little difficulty for model training. The main challenge lies in the implicit estimation of the **direct-path time shift**, which causes a misalignment between the reverberant input and the clean target. Hence, we propose using time-shifted anechoic clean speech as the learning target, which effectively addresses this issue and significantly improves both speech quality and ASR accuracy. Although previous studies (Delfarah et al., 2020; Zhao et al., 2020; Wang et al., 2021) have also used this as a learning target, they did not compare the results across different targets.

Model architecture: In this Challenge, Sun et al. (2025) proposed a channel-mixing module between time and frequency modeling, along with progressive block extension to enable model training at different scales. Although it achieves strong performance across different metrics, some over-smoothing artifacts can be observed under bandwidth limitation and packet loss (Saijo et al., 2025), as it relies solely on a regression model. Other works have explored integrating the outputs of regression and generative models in different ways. Chao et al. (2025) combines the outputs using a simple energy criterion based on the noisy input. Both Rong et al. (2025) and Goswami & Harada (2025) propose three-stage frameworks for generating the final USE results. Rong et al. (2025) employs filling, separation, and restoration modules, while Goswami & Harada (2025) uses a fusion network to combine the outputs of the regression and token sampling–based generative models. Le et al. (2025) applies a four-stage strategy consisting of audio declipping, packet loss compensation, audio separation, and spectral inpainting modules.

Under severe degradation of input speech, regression models preserve fidelity but yield over-smoothed outputs, whereas generative models keep good quality but risk hallucination (see Section 2.2 for a detailed discussion). Effectively combining the strengths of both approaches to achieve high quality while preserving fidelity thus becomes critically important. Guided by theoretical insights, we adopt a simple two-stage framework: first, the regression model is trained to convergence and its parameters are frozen; then, its outputs are used as conditional inputs to the generative model. Our experimental results show that this approach strikes a good balance between quality and fidelity.

Training data quality: Since deep learning models are data-driven, both the quality and quantity of training data play a critical role in their performance. While prior studies (Zhang et al., 2024a; Gonzalez et al., 2024) have examined the impact of training data scale on speech enhancement models, investigations into the effect of data quality remain relatively scarce. To the best of our knowledge, only a recent study (Li et al., 2025) has highlighted that, within large-scale training sets, prioritizing high-quality data is more important. In this paper, we discuss the trade-off between training data scale and quality. We found that although the URGENT Challenge organizers have already filtered the training data, many remaining samples still contain significant degradations, not only due to the use of early-reflected speech. Including such degraded samples into the training data also imposes an upper bound on USE performance when addressing common background noise, such as electrical microphone hiss (see Section 3.8 for a detailed discussion).

## 2 PROPOSED METHOD

In this section, we provide a detailed discussion of the common challenges in universal speech enhancement and present our proposed methods to address them. We first present our findings and solutions for using anechoic clean speech (without early reflections) as the learning target in the speech dereverberation task. Next, we propose a strategy to effectively combine generative and regression models, aiming to preserve fidelity while mitigating the over-smoothing problem. Finally, we present our approach to investigating the trade-off between training data scale and quality, a relatively new topic that has also been explored in a recent study (Li et al., 2025).

### 2.1 CHALLENGE AND SOLUTION TO APPLY ANECHOIC CLEAN SPEECH AS A LEARNING TARGET FOR SPEECH DEREVERBERATION

Given an anechoic clean speech $s$ and a RIR $r$, the reverberant speech $y$ can be modeled as convolution between $s$ and $r$ as follows:

$$y[n] = s[n] * r[n] \qquad (1)$$

where $n$ denotes the discrete time index, $*$ denotes the convolution operator. The RIR $r[n]$ can be further decomposed into direct path $\delta[n - n_0]$, early reflection $r_e[n]$, and late reflection $r_l[n]$ (see Figure 4 in the Appendix as a reference):

$$r[n] = \delta[n - n_0] * (r_e[n] + r_l[n]) \qquad (2)$$

In this formulation, $\delta[n]$ denotes the Dirac delta function, and $n_0$ represents the time shift introduced by the direct path, typically ranging from 5 to 30 ms. We estimate $n_0$ based on the maximum magnitude of the RIR, i.e., $n_0 = \arg\max_n |r[n]|$. Early reflections are generally defined as RIR components occurring within 50 ms after the direct-path peak $\delta[n - n_0]$, as specified in the URGENT Challenge (Zhang et al., 2024b; Saijo et al., 2025) and other studies (Wang et al., 2021). The remaining impulses are treated as late reflections (Naylor & Gaubitch, 2010).

When training a speech dereverberation model, instead of using anechoic clean speech $s$ as learning targets, most previous works, including the previous URGENT Challenges, employ early reflected speech $s_e[n] = s[n] * \delta[n - n_0] * r_e[n]$ as the target, because "early reflections are much harder to remove, and the difficulty of solving the problem leads to excessive artifacts in the enhanced speech" (Valin et al., 2022; Zhou et al., 2023; Zhao et al., 2020). To examine this issue, we first used anechoic clean speech $s$ directly as the learning target, but this resulted in the poorest performance, consistent with findings from previous studies.

Although early reflections have a smaller impact on perceived quality compared to late reverberation (Valin et al., 2022), we found that they still cause significant degradation in speech quality as estimated using different quality metrics (e.g., UTMOS (Saeki et al., 2022), DNSMOS (Reddy et al., 2022), and NISQA (Mittag et al., 2021)), and would therefore affect the speech enhancement quality when they're treated as the learning target. For example, when we gradually reduce the early reflection window from 50 ms to 10 ms and then to 0 ms, we observe a corresponding improvement in the quality score of the enhanced speech, as shown in Figure 3 (b).

In the case of setting early reflection window as 0 ms, the learning target simply becomes the time-shifted version of $s$, i.e., $s[n] * \delta[n - n_0] = s[n - n_0]$. Consequently, in Equation 2, eliminating early reflections poses little difficulty for the model. The critical challenge is the implicit estimation of the **direct-path time shift**, which introduces a misalignment of $n_0$ between the reverberant input and the clean target. This explains why using clean speech $s[n]$ directly as the learning target produced the worst results, whereas $s[n - n_0]$ yielded the best quality, and the early-reflection target ($s_e[n] = s[n] * \delta[n - n_0] * r_e[n] = s[n - n_0] * r_e[n]$) adopted in prior studies proved sub-optimal.

## 2.2 Effective Combination of Generative and Regression Models

In universal speech enhancement, the goal is to restore the quality of degraded speech while preserving **fidelity**, ensuring that all other factors remain unchanged, e.g., linguistic content, speaker identity, emotion, accent, and other paralinguistic attributes. Generative and regression (also called discriminative) models tackle the problem in distinct ways. The regression model outputs the conditional expectation, $\hat{s} = E[S \mid Y = y]$, whereas the generative model produces samples from the conditional distribution, $\hat{s} \sim p(s \mid y)$, where $y$ corresponds to the degraded input speech signal. When $y$ is severely degraded (i.e., contains little information about the true clean signal, for example, packet loss, bandwidth limitation, low-SNR conditions, etc.), the regression output is biased towards prior mean $E[S \mid Y = y] \approx E[S]$. If the prior is multimodal (e.g., many plausible phonemes), the mean may be a blend of modes, which sounds muffled and unnatural, resulting in **over-smoothing problem** (Ren et al., 2022; Chao et al., 2025). Conversely, in such a scenario, the generative model produces outputs drawn from the prior distribution, $p(s \mid y) \approx p(s)$, enabling the generation of natural-sounding speech consistent with the prior. Nevertheless, the lexical/linguistic content or speaker characteristics are not guaranteed to match the original signal, resulting in **hallucination problem** as defined in (Saijo et al., 2025; Scheibler et al., 2024). In summary, when $y$ is less informative, the regression model excels at preserving fidelity but may fail to improve quality, whereas the generative model enhances quality but often struggles to preserve fidelity. As demonstrated in (Blau & Michaeli, 2018), a fundamental trade-off exists between fidelity and perception. Specifically, posterior sampling from the posterior $p(s \mid y)$ results in a mean squared error (MSE) that is twice the minimum MSE (MMSE), which is achieved by the regression model (Blau & Michaeli, 2018).

Considering the applications of universal speech enhancement, we argue that preserving fidelity—particularly the linguistic content—is of greater importance. Therefore, our strategy is to first employ a regression model to estimate the posterior mean, and then leverage a generative model to **only correct** the over-smoothed regions. As shown in Figure 1, the regression model is first trained until convergence, after which its parameters are frozen. As inspired by DeepFilterGAN (Serbest et al., 2025), the model's outputs, along with the noisy input, are subsequently used as inputs to the generative model. Incorporating a residual connection between the regression model output and the final output enables the generative model to focus primarily on regions whose characteristics diverge from those of real data.

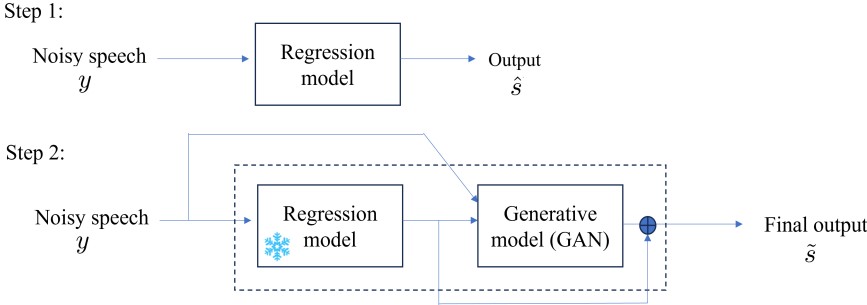

Figure 1: The two-stage framework that combines a regression model and a generative model.

In the following, we offer some theoretical justification for this model combination. A recent finding (Freirich et al., 2021) indicates that the optimal model, which minimizes MSE while satisfying the constraint of perfect perception, can be derived by **optimally transporting** the posterior mean prediction (i.e., the MMSE estimate) to the true data distribution. Ohayon et al. (2025) approximate the optimal transport using flow matching and achieve improved MSE in image restoration. In addition to speech enhancement, SEStream (Huang et al., 2023) and StoRM (Lemercier et al., 2023) also achieve strong results in codec compression and dereverberation, respectively, by using a regression model to provide an initial prediction for a subsequent generative model.

In this study, we leverage GAN-based methods to approximate optimal transport. Wasserstein GANs (Arjovsky et al., 2017) optimize an objective equivalent to the Wasserstein-1 distance between the source and target distributions, a principled metric from optimal transport theory. Furthermore, a single forward-pass generation in GAN-based methods makes them more easily adaptable to real-time scenarios. In addition, because a convolutional neural network (CNN) is used as the discriminator, each element in the final feature map (prior to the averaging operation for the final prediction) has only a limited receptive field. Assuming the discriminator is Lipschitz continuous (e.g., via spectral normalization (Miyato et al., 2018)), the following constraint holds:

$$\|D^{(l)}(\tilde{s}) - D^{(l)}(s)\| \leq L\|\tilde{s} - s\|, \quad \forall l \tag{3}$$

where $D(.)$ is the discriminator, $l$ is the index of the discriminator layer, $L$ is the Lipschitz constant and $\tilde{s}$ is the final model output. Here, let us focus on the receptive field of one element in the final feature map. When the region $\tilde{s}$ closely approximates the target clean speech ($\|\tilde{s} - s\| \approx 0$), the left-hand side of Equation 3—representing the **feature-matching loss** during generator training—converges toward 0. Therefore, the generative model can mainly focus on correcting the over-smoothed regions of the regression model output. Consequently, this two-stage framework can keep the fidelity while improving speech quality.

### 2.3 TRADE-OFF BETWEEN TRAINING DATA SCALE AND QUALITY

In Track 1 of the URGENT 2025 Challenge, the training dataset comprises of 2,500 hours of speech from various sources: CommonVoice ($\sim$ 1,300 hours)(Ardila et al., 2020), DNS5 ($\sim$ 350 hours)(Dubey et al., 2024), MLS ($\sim$ 450 hours)(Pratap et al., 2020), LibriTTS ($\sim$ 200 hours)(Zen et al., 2019), VCTK ($\sim$ 80 hours)(Veaux et al., 2013), WSJ ($\sim$ 85 hours)(Garofolo et al., 2007),

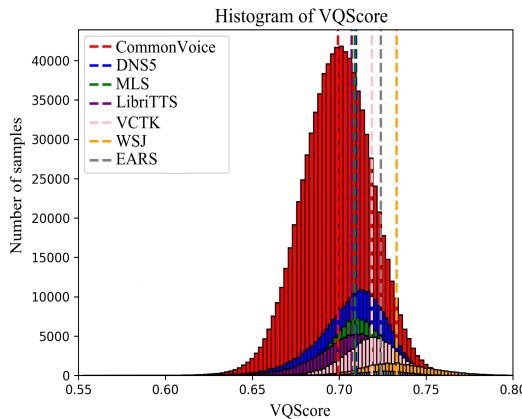

Figure 2: Histogram of VQScore for speech sources in each subset of the URGENT 2025 Challenge Track 1. The median of each data source is indicated by a dashed vertical line.

and EARS (∼ 85 hours) (Richter et al., 2024). Although the organizers have already filtered out non-speech samples using voice activity detection (VAD) and removed noisy samples based on the DNSMOS score, many recordings with audible background noise remain even after this filtering (Saijo et al., 2025). Leveraging its high correlation with subjective scores (Zhang et al., 2025b) and fast inference capability (processing 2,500 hours of speech in less than 8 hours on a single NVIDIA A100 GPU), we employed VQScore (Fu et al., 2024) to analyze the quality distribution of each training data source. Figure 2 illustrates the VQScore distribution for each speech source (individual source histograms are provided in Figure 5 in the Appendix). The figure shows that, despite comprising the majority of the dataset, CommonVoice exhibits the lowest quality, reflecting its crowdsourced nature. In contrast, WSJ, EARS, and VCTK consistently demonstrate the highest quality on average. Upon examining several low-quality examples from different sources, we found that most contained either stationary background noise or consisted entirely of non-speech noise. These training targets may confuse the speech enhancement model and degrade its performance. Therefore, a threshold on VQScore is applied to further discard such samples. We also observed that some speech samples from the EARS dataset received low quality ratings because they were expressive (e.g., whispering, high-pitched), which can be considered out-of-domain for the evaluation model. Overall, the speech in the EARS dataset is the cleanest, and we leverage this characteristic to further fine-tune our model (see Section 3.8). Examples of low VQScore samples from each source are provided in the supplementary material.

## 3 Experiments

### 3.1 Dataset

As noted earlier, the URGENT 2025 Challenge training dataset comprises multi-condition speech recordings across five languages (English, German, French, Spanish, and Chinese) with diverse sampling frequencies (8, 16, 22.05, 24, 32, 44.1, and 48 kHz), along with noise samples and RIRs, as summarized in Table 4 in the Appendix. Seven types of distortions are considered: additive noise, reverberation, clipping, bandwidth limitation, codec artifacts, packet loss, and wind noise. We followed the organizers' guidelines to simulate the validation set using the validation splits of the corpora (Saijo et al., 2025). The non-blind test set of URGENT 2025, consisting of 1,000 utterances with noise and RIRs from unseen sources, is used to evaluate the models.

### 3.2 Model Architecture

To enable a single model to operate across different sampling rates, we employ sampling frequency-independent (SFI) STFT (Zhang et al., 2023a), which dynamically adjusts the FFT window and hop size according to the input sampling rate, ensuring a fixed time duration and consistent feature frame length across all sampling rates. To ensure that inputs with different sampling rates yield an integer

number of frequency bins, we set the FFT window size to 320 points for the 8 kHz case (i.e., for all sampling frequencies, we use a 40 ms window size and a 5 ms hop size.).

Since the model architecture is not the focus of this paper, we adopt USEMamba with 30 layers (Chao et al., 2025; 2024) as the regression model. USEMamba alternates between two types of sequence modeling modules (i.e., Mamba) applied to frequency features and time features in the TF domain. For GAN training, the generator is a 6-layer USEMamba, while the discriminators are CNN-based.

To account for distinct feature patterns across frequency bands and support speech with varying sampling rates, we propose an **adaptive multi-band discriminator**, inspired by Kumar et al. (2023). For each band—corresponding to the input sampling rate (e.g., for 8 kHz, only one sub-band from 0–4 kHz; for 22.05 kHz, three sub-bands: 0–4 kHz, 4–8 kHz, and 8–11.025 kHz)—a 5-layer 2-D convolutional network is employed for feature extraction. The sub-band features are concatenated along the frequency axis and passed through a final 2-layer 2-D convolution with global average pooling to produce the discriminator output. The models are trained on 8 NVIDIA A100 GPUs with a batch size of 1, allowing longer input speech to be processed without memory issues. All models (i.e., the regression model, generator, and discriminator in the GAN) are optimized using AdamW with a learning rate of 0.0002. To ensure reproducibility, the code will be made publicly available upon acceptance of the paper.

### 3.3 EVALUATION METRICS

Given the dual objectives of speech enhancement—improving perceptual quality while maintaining fidelity—a comprehensive evaluation requires multiple metrics. For fidelity and quality, Perceptual Evaluation of Speech Quality (PESQ) measures perceptual quality (Rix et al., 2001), Extended Short-Time Objective Intelligibility (ESTOI) evaluates intelligibility (Jensen & Taal, 2016), and signal-to-distortion ratio (SDR) quantifies waveform distortion in the time domain (Roux et al., 2019). Mel Cepstral Distortion (MCD) and Log-Spectral Distance (LSD) capture spectral deviations between enhanced and reference speech. Additionally, results on downstream tasks were also evaluated. Task-independent metrics such as SpeechBERTScore (SBERT)(Saeki et al., 2024), which quantifies enhancement quality using a self-supervised model, and Levenshtein Phoneme Similarity (LPS)(Pirklbauer et al., 2023), which measures phoneme sequence similarity; and task-dependent metrics such as speaker similarity (SpkSim), which reflects preservation of speaker identity, and character accuracy (CAcc), which reflects ASR performance. Finally, non-intrusive metrics—including DNSMOS (Reddy et al., 2022), NISQA (Mittag et al., 2021), and UTMOS (Saeki et al., 2022)—estimate perceptual quality without requiring a clean reference, but focus only on quality rather than fidelity.

Note that, except for the non-intrusive metrics and CAcc, all evaluation metrics require clean speech as a reference. In the speech dereverberation task, however, the organizers provide **early-reflected speech** as the clean reference. **This mismatch implies that our model—trained using time-shifted anechoic clean speech targets—may receive lower leaderboard scores on PESQ, ESTOI, SDR, MCD, LSD, SBERT, LPS, and SpkSim, even if the actual perceptual quality and fidelity are improved.**

### 3.4 RESULTS ON TRAINING DATA FILTERING BASED ON THE QUALITY ESTIMATION

Based on the observations in Section 2.3, we first aim to determine a suitable VQScore threshold, such that training samples with scores below the threshold are excluded. Three thresholds are considered: 0.50 (no filtering), 0.65, and 0.72, corresponding to 2,518 (original size), 2,506, and 629 hours of training data, respectively. We then train three models on these datasets using time-shifted anechoic clean speech as targets and present their UTMOS learning curves on the validation set in Figure 3 (a). We observe that without VQScore filtering (blue line), the model performs the worst. With a threshold of 0.72 (green line), the model achieves the best performance in the early stage of training due to the higher quality of data. However, in the later stage, its performance lags behind the model trained with a threshold of 0.65 (orange line), likely due to the limited amount of training data. Based on this finding, in the following experiments, unless noted otherwise, we adopt a threshold of 0.65, as it strikes a good balance between training data quality and quantity.

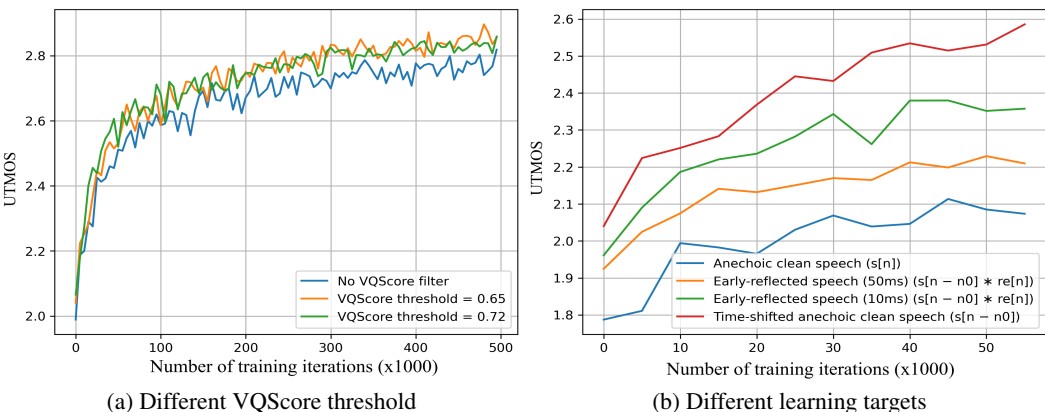

(a) Different VQScore threshold    (b) Different learning targets

Figure 3: Learning curves of UTMOS scores on the validation set under (a) different VQScore filtering thresholds, and (b) different learning targets.

Table 1: Non-blind test set results of the URGENT 2025 Challenge. All metrics are "higher is better", except MCD and LSD. Rank $N$ denotes the system ranked $N^{th}$ in the challenge. Note that shaded metrics are not directly comparable across the two learning targets due to mismatches in the definition of the "clean" reference.

| Team / | Non-intrusive SE metrics | | | Intrusive SE metrics | | | | | Task-ind. | | Task-dep. | |
|---|---|---|---|---|---|---|---|---|---|---|---|---|
| Rank | DNSMOS | NISQA | UTMOS | PESQ | ESTOI | SDR | MCD↓ | LSD↓ | SBERT | LPS | SpkSim | CAcc |
| Noisy | 1.84 | 1.69 | 1.56 | 1.37 | 0.61 | 2.53 | 7.92 | 5.51 | 0.75 | 0.62 | 0.63 | 81.29 |
| Baseline | 2.94 | 2.89 | 2.11 | 2.43 | 0.80 | 11.29 | 3.32 | 2.85 | 0.86 | 0.79 | 0.80 | 84.96 |
| Rank 3 | 3.00 | 3.45 | 2.31 | 2.74 | 0.84 | 13.06 | 3.30 | 3.08 | 0.89 | 0.84 | 0.83 | 87.94 |
| Rank 2 | 3.01 | 3.21 | 2.30 | 2.79 | 0.85 | 13.11 | 2.93 | 2.94 | 0.90 | 0.85 | 0.84 | 88.05 |
| Rank 1 | 3.01 | 3.41 | 2.40 | **2.95** | **0.86** | **14.33** | 3.01 | 2.83 | **0.91** | **0.86** | 0.85 | 88.92 |
| **Early reflected** | 3.06 | 3.23 | 2.26 | 2.81 | 0.85 | 12.28 | **2.87** | **2.66** | 0.90 | 0.84 | 0.82 | 87.62 |
| **+GAN correction** | 3.04 | 3.53 | 2.30 | 2.78 | 0.84 | 12.25 | 2.97 | 2.75 | 0.90 | 0.84 | **0.85** | 88.13 |
| **Shifted anechoic** | 3.25 | 3.85 | 2.76 | 2.41 | 0.77 | 8.23 | 3.63 | 3.28 | 0.89 | 0.84 | 0.82 | 89.41 |
| **+GAN correction** | **3.26** | **4.12** | **2.80** | 2.38 | 0.76 | 8.18 | 3.73 | 3.51 | 0.89 | 0.84 | 0.83 | **89.88** |

## 3.5 RESULTS OF APPLYING TIME-SHIFTED ANECHOIC CLEAN SPEECH AS TARGETS

In this section, we first present the UTMOS learning curves under different training targets, shown in Figure 3 (b). As discussed in Section 2.1, directly using anechoic clean speech $s[n]$ as the learning target yields the worst performance, consistent with previous studies. Hence, in the following experiment, we ignore this learning target. In Table 1, we then present the results of our proposed approach, which uses time-shifted anechoic clean speech ($s[n - n_0]$) as the learning target for speech dereverberation. The table also presents the performance of the baseline (TF-GridNet) (Wang et al., 2023) and the top three ranked systems ($1^{st}$: Team Bobbsun (Sun et al., 2025), $2^{nd}$: Team rc (Chao et al., 2025), and $3^{rd}$: Team Xiaobin (Rong et al., 2025)) as a reference. The full results are available on the leaderboard [1]. From the table, it is clear that replacing early-reflected speech with time-shifted anechoic clean targets yields substantial improvements on non-intrusive quality metrics (DNSMOS from 3.06 to 3.25, NISQA from 3.23 to 3.85, and UTMOS from 2.26 to 2.76), and ASR performance (CAcc from 87.62 to 89.41). The improvement in CAcc implicitly verifies that the quality gains do not come at the expense of hallucination. As noted in Section 3.3, the remaining metrics may not accurately reflect the performance of models trained with shifted anechoic speech, as the evaluation reference on the leaderboard is based on early-reflected speech. To address this issue, Table 5 presents the evaluation results computed using anechoic clean speech as the reference. From the table, we can observe that using shifted anechoic speech as the learning target significantly outperforms using early-reflected speech. . These results indicate that using early-reflected speech as a learning target still degrades the output quality of the USE model, as the enhanced audio sounds more **reverberant**. This is further illustrated by the spectrogram comparison in Figure 6 in the Appendix. Some audio samples are provided in the demo page.

---

[1] https://urgent-challenge.github.io/urgent2025/leaderboard/

Table 2: Comparison with other open-source USE models on the subsets of the URGENT 2025 Non-Blind Test Set. All metrics are "higher is better," except MCD and LSD.

| Team / Rank | Non-intrusive SE metrics | | | Intrusive SE metrics | | | | | Task-ind. | | Task-dep. | |
|---|---|---|---|---|---|---|---|---|---|---|---|---|
| | DNSMOS | NISQA | UTMOS | PESQ | ESTOI | SDR | MCD↓ | LSD↓ | SBERT | LPS | SpkSim | CAcc |
| 48k | | | | | | | | | | | | |
| Noisy | 2.04 | 1.83 | 1.99 | 1.28 | 0.56 | 2.29 | 7.77 | 5.50 | 0.78 | 0.77 | 0.69 | 90.60 |
| ClearerVoice | 2.97 | 3.38 | 3.02 | 2.09 | 0.72 | 11.55 | 5.08 | 5.15 | 0.85 | 0.87 | 0.63 | 89.90 |
| Proposed | 3.31 | 4.41 | 3.55 | 2.65 | 0.77 | 12.79 | 3.93 | 3.19 | 0.89 | 0.91 | 0.87 | 92.50 |
| 44.1k | | | | | | | | | | | | |
| Noisy | 1.91 | 1.79 | 1.52 | 1.33 | 0.64 | 3.34 | 7.44 | 5.62 | 0.76 | 0.63 | 0.71 | 83.60 |
| Resemble Enhance | 3.13 | 3.68 | 2.11 | 1.33 | 0.45 | -15.01 | 11.02 | 7.93 | 0.69 | 0.47 | 0.61 | 47.20 |
| Proposed | 3.32 | 4.15 | 2.68 | 2.28 | 0.78 | 7.18 | 3.91 | 3.47 | 0.90 | 0.85 | 0.88 | 92.20 |

## 3.6 RESULTS OF COMBINING GENERATIVE AND REGRESSION MODELS

To verify that the two-stage framework allows the GAN to focus on correcting over-smoothed regions while ignoring well-predicted parts of the regression output, we compute the average Spearman's rank correlation coefficient between the magnitude spectrogram of the clean-regression residual (clean speech ($s$) - regression output ($\hat{s}$)) and the final-regression residual (final output ($\tilde{s}$) - regression output ($\hat{s}$)) on the non-blind set, obtaining a high correlation of 0.78. The theoretical rationale is that, as shown in Equation 3, the generator's feature matching loss is bounded above by the distance between the regression output and the clean speech. From Table 1, it can be observed that applying GAN to refine the regression output significantly improves NISQA, SpkSim, and CAcc, with moderate gains in UTMOS. In addition, it shows only a marginal or negligible impact on most intrusive SE metrics, indicating that fidelity is well preserved. The overall ranking table for the non-blind test set is provided on the demo page, where we can see that applying GAN correction leads to a higher ranking. To further compare our two-stage framework with other GAN training strategies, Figure 7 presents the learning-curve comparison on the validation set between the common approach (pre-training with a regression loss followed by adversarial fine-tuning) and our two-stage GAN correction. Throughout the entire training process, the two-stage framework consistently achieves lower magnitude, phase, and time-domain losses, as well as higher PESQ scores. **In addition, our shifted anechoic target with GAN correction achieves state-of-the-art performance across all non-intrusive metrics and ASR CAcc.**

## 3.7 COMPARISON WITH OTHER OPEN-SOURCE USE MODELS

In this section, we compare our proposed USE model (shifted anechoic + GAN correction) with other popular open-source USE models, namely ClearerVoice-Studio (Zhao et al., 2025)[2] and Resemble Enhance[3] . While this comparison is not entirely fair due to differences in training data, it still provides useful insights into the relative strengths and limitations of existing approaches. Since ClearerVoice-Studio only supports 16 kHz or 48 kHz inputs, and Resemble Enhance only supports 44.1 kHz, we report enhancement results on the corresponding subsets of the non-blind test set from the URGENT 2025 Challenge, as shown in Table 2. Although our proposed method outperforms ClearerVoice-Studio, a regression-based model built on MossFormer2 (Zhao et al., 2024), the latter still demonstrates reasonable enhancement performance. On the other hand, while Resemble Enhance (based on latent conditional flow matching) improves non-intrusive quality metrics, its low intrusive metrics and CAcc results suggest that it may hallucinate content—a common issue with purely generative models (Saijo et al., 2025).

## 3.8 EVALUATION ON UNSEEN LANGUAGES

One emerging application of USE is training data cleaning for other speech generative models (e.g., text-to-speech, TTS) (Koizumi et al., 2023c; Karita et al., 2025; Koizumi et al., 2023b; Ma et al., 2024). This is particularly important for low-resource languages, where studio-quality speech data is often scarce. To support this goal, the USE model must be **language-agnostic**. Saijo et al. (2025) shows that regression models are relatively insensitive to language variations, whereas purely

---

[2]https://github.com/modelscope/ClearerVoice-Studio
[3]https://github.com/resemble-ai/resemble-enhance

Table 3: Speech enhancement results for unseen languages from the FLEURS dataset.

|  | DNSMOS | SpkSim | CAcc |
|---|---|---|---|
| Italian (it_it) | | | |
| Original | 3.12 | - | 97.28 |
| FLEURS-R  (Ma et al., 2024) | **3.37** | 0.87 | 97.69 |
| Proposed | 3.20 | **0.98** | 97.00 |
| Proposed (EARS) | 3.27 | 0.97 | **98.09** |
| Dutch (nl_nl) | | | |
| Original | 2.99 | - | **97.40** |
| FLEURS-R  (Ma et al., 2024) | **3.36** | 0.88 | 97.19 |
| Proposed | 3.13 | **0.97** | 97.18 |
| Proposed (EARS) | 3.28 | 0.95 | 97.26 |
| Japanese (ja_jp) | | | |
| Original | 2.96 | - | 95.34 |
| FLEURS-R  (Ma et al., 2024) | **3.36** | 0.88 | 95.14 |
| Proposed | 3.07 | **0.98** | 95.30 |
| Proposed (EARS) | 3.18 | 0.95 | **95.43** |

generative models (e.g., latent diffusion and vocoding) exhibit a strong dependency on language. This is also one of the reasons we adopt a two-stage framework, where the generative model is used solely to refine the over-smoothed regions of the regression model's output.

Following Miipher-2  (Karita et al., 2025), we use the FLEURS dataset  (Conneau et al., 2023) to evaluate model performance on unseen languages. Specifically, we evaluate three languages (Italian, Dutch, and Japanese), assessing speech quality with DNSMOS, speaker similarity (compared to the original)  (Desplanques et al., 2020), and ASR  (Radford et al., 2023) CAcc before and after different speech restoration models, with the results summarized in Table 3. The FLEURS-R  (Ma et al., 2024) speech was restored using the Miipher-2 model and obtained from the official source[4]. The model's acoustic feature extractor leverages the Universal Speech Model (Zhang et al., 2023b), which was pre-trained on 12 million hours of speech spanning more than 300 languages. The enhanced features are subsequently converted back to a waveform using the WaveFit vocoder  (Koizumi et al., 2023a).

We observed that some samples in the FLEURS dataset contain only slight stationary wideband noise or electrical microphone hiss (see Figure 8 in the Appendix)—artifacts that can still appear in our training data even after VQScore filtering. Our original model (shifted anechoic + GAN) cannot effectively remove such background noise, as some 'clean' training examples encourage the model to retain these distortions. To address this, we fine-tune the model using only one of the highest-quality subsets of our training data (the EARS dataset), with the results reported as Proposed (EARS) in Table 3. From the table, we observe that although FLEURS-R achieves slightly higher speech quality, its speaker characteristics deviate more from the original speech compared to our proposed methods. This suggests that relying solely on generative restoration may enhance perceptual quality at the expense of speaker fidelity, highlighting the importance of balancing the two objectives. Between the two proposed methods, Proposed (EARS) achieves better speech quality and accuracy of ASR, further underscoring the importance of training data quality.

## 4 CONCLUSION

This paper systematically investigates three key challenges in developing a universal speech enhancement (USE) model. First, we establish the importance of training targets by showing that time-shifted anechoic clean speech—rather than conventional early-reflected speech—significantly improves both perceptual quality and downstream ASR performance. Second, to balance fidelity and perceptual quality, we provide a theoretical analysis and propose a two-stage framework that

---

[4]https://huggingface.co/datasets/google/fleurs-r

integrates a regression model for fidelity with a generative model for perceptual refinement, ensuring that the generative model only corrects over-smoothed regions without hallucinating content. Third, we underscore the critical role of training data quality, revealing a trade-off between scale and cleanliness. In particular, we demonstrate that fine-tuning on the cleanest available subset yields further improvements in enhanced speech quality. From an application perspective, the proposed USE model exhibits strong language-agnostic and high-fidelity capabilities, making it well-suited for improving the quality of training data for downstream speech generative tasks.

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

# A    APPENDIX

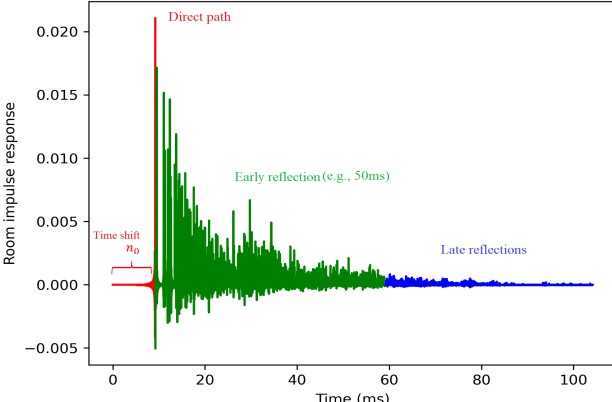

Figure 4: An example of a room impulse response, highlighting the time shift $n_0$ introduced by the direct path.

Table 4: Dataset Composition for URGENT 2025 Challenge

| Type | Corpus | Condition | Sampling (kHz) | Duration (h) |
|------|--------|-----------|----------------|--------------|
| Speech | LibriVox (DNS5) | Audiobook | 8–48 | 350 |
| | LibriTTS | Audiobook | 8–24 | 200 |
| | VCTK | Newspaper-style | 48 | 80 |
| | WSJ | WSJ news | 16 | 85 |
| | EARS | Studio recording | 48 | 100 |
| | Multilingual Librispeech (de, en, es, fr) | Audiobook | 8–48 | 450 |
| | CommonVoice 19.0 (de, en, es, fr, zh-CN) | Crowd-sourced voices | 8–48 | 1300 |
| Noise | AudioSet+FreeSound (DNS5) | Crowd-sourced + YouTube | 8–48 | 180 |
| | WHAM! Noise | 4 urban environments | 48 | 70 |
| | FSD50K (Filtered) | Crowd-sourced | 8–48 | 100 |
| | Free Music Archive | Directed by WFMU | 8–44.1 | 200 |
| RIR | Simulated RIRs (DNS5) | SLR28 | 48 | 60k samples |

Table 5: Non-blind test set results for the URGENT 2025 Challenge, referenced against **anechoic clean speech**.

| Method | PESQ | ESTOI | SBERT | LPS | SpkSim |
|--------|------|-------|-------|-----|--------|
| Early reflected + GAN | 2.40 | 0.69 | 0.88 | 0.83 | 0.83 |
| Shifted anechoic + GAN | **2.71** | **0.78** | **0.89** | **0.86** | **0.84** |

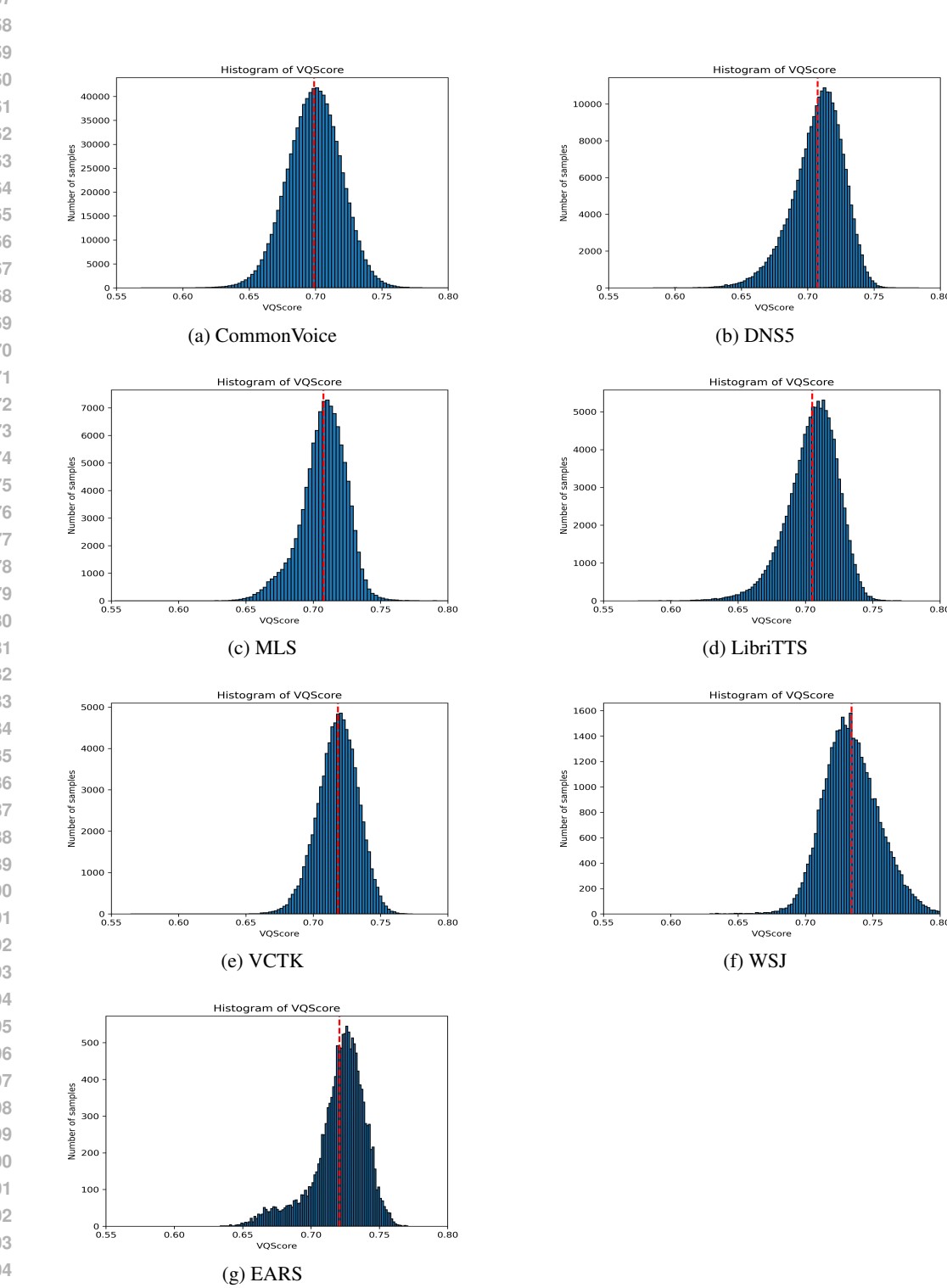

Figure 5: Histogram of VQScore across different speech sources in the URGENT 2025 Challenge Track 1. The median of each data source is indicated by a dashed vertical line.

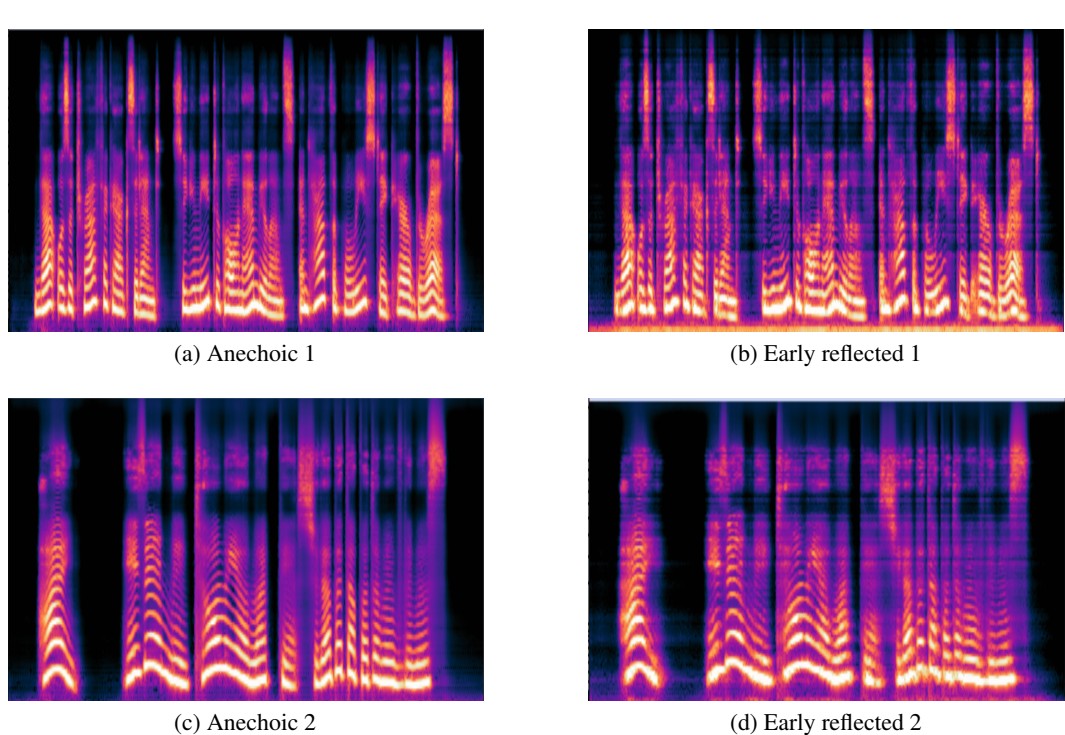

(a) Anechoic 1

(b) Early reflected 1

(c) Anechoic 2

(d) Early reflected 2

Figure 6: Enhanced spectrogram comparison between using time-shifted anechoic clean speech and early-reflected speech as learning targets. (a) and (b) correspond to the same noisy input, and (c) and (d) correspond to another noisy input. Both samples are drawn from the blind-test set.

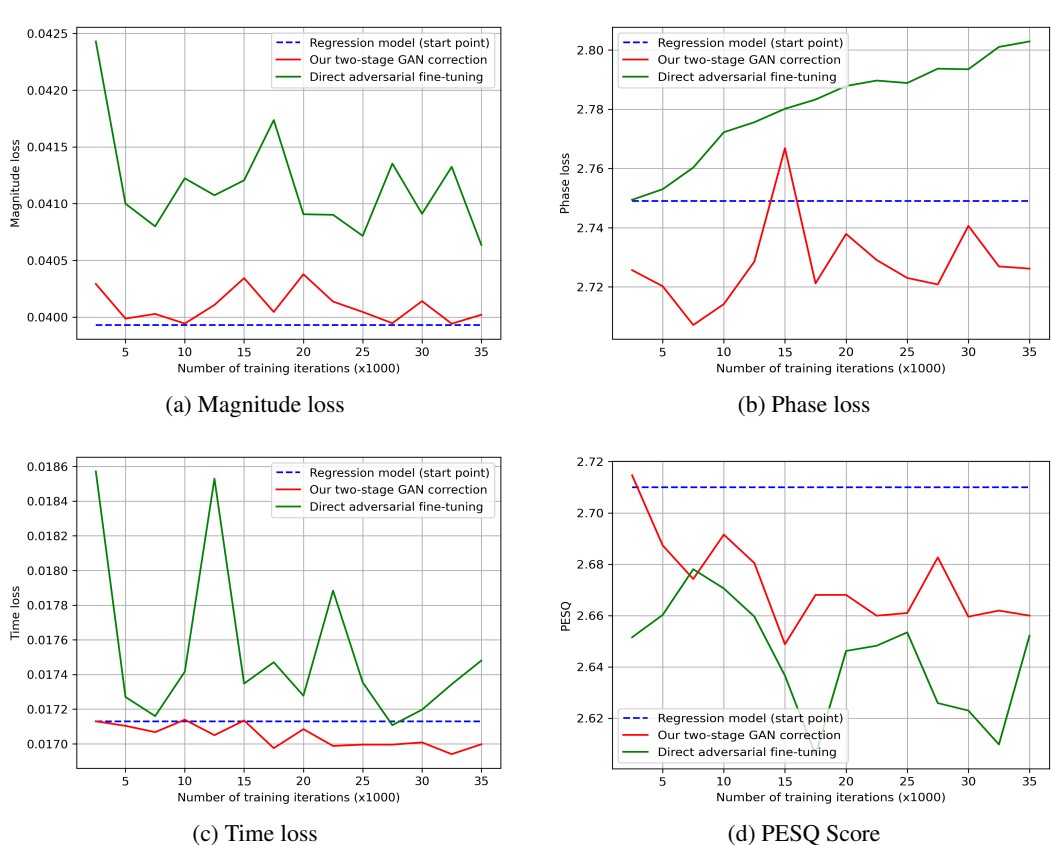

(a) Magnitude loss

(b) Phase loss

(c) Time loss

(d) PESQ Score

Figure 7: Learning curves comparison on validation-set between pre-training with a regression loss followed by adversarial fine-tuning and our two-stage GAN correction. (a) Magnitude loss, (b) Phase loss, (c) Time loss, and (d) PESQ score.

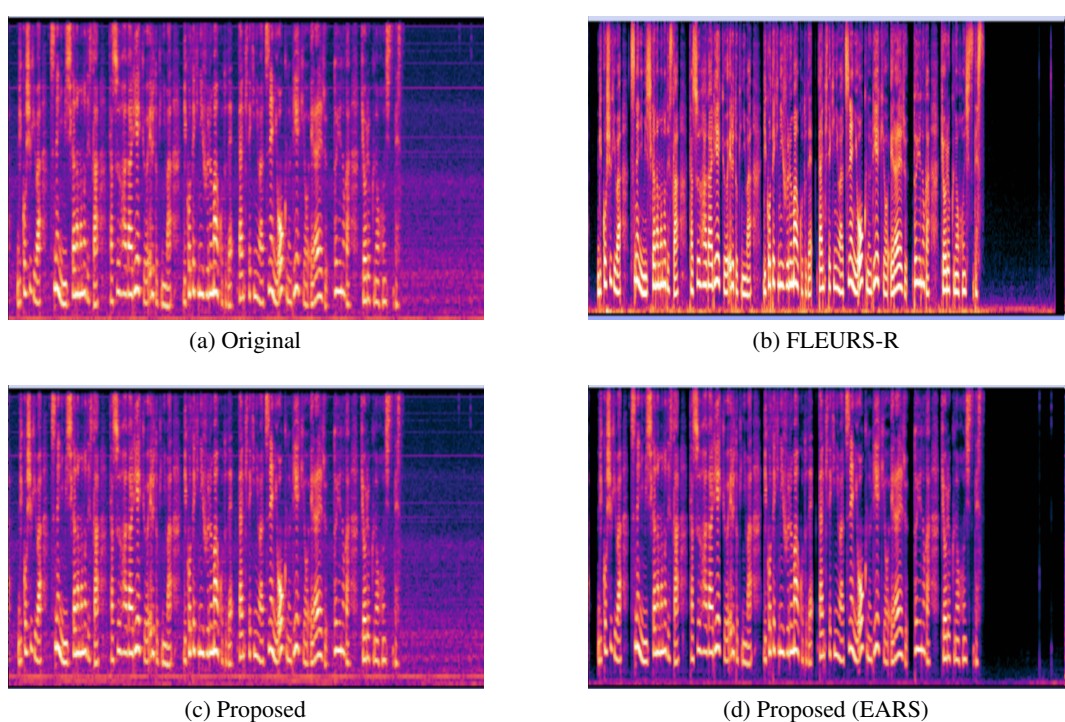

(a) Original       (b) FLEURS-R

(c) Proposed       (d) Proposed (EARS)

Figure 8: Spectrogram comparison of a Japanese utterance (9997427445140542468.wav) from the FLEURS dataset. The original speech contains some stationary wideband noise.

