# OpenReview forum: "You Are What You Train: Rethinking Training Data Quality, Targets, and Architectures for Universal Speech Enhancement"
_ICLR.cc/2026/Conference — Submitted to ICLR 2026_

### Official Review · Reviewer_CvYq · 2025-10-29

**Soundness:** 1
**Presentation:** 2
**Contribution:** 1
**Rating:** 2
**Confidence:** 4

**Summary:**

The paper investigates the causes of performance degradation in universal speech enhancement (SE) methods. It considers alignment mismatch between input and target recordings, compares regression-based and generative approaches to SE, and investigates the effects of low-quality training data on the final model's performance.

**Strengths:**

The paper correctly identifies key areas that crucially affect the performance of SE methods.

**Weaknesses:**

The paper has the following weaknesses:
1. The paper claims that aligning target and input audio recordings improves the overall performance of the model. This is not a new observation; e.g. [2] also aligns target and input audios for training. Moreover, this observation is quite obvious, since any impulse response used for augmentation introduces a time shift that is known and can be manually corrected relatively simply. Therefore, the observation that using $s[n-n_0] = s[n] \ast \delta[n - n_0]$ yields better results is not a novelty.

2. The authors discuss the benefits and downsides of using regression-based and generative methods for SE. They argue that using generative modelling can help reduce over-smoothing, yet preserve the fidelity. **Firstly**, the problem of over-smoothing is already well-studied, among others, in [1, 2, 3]. Other GAN-based methods deal with over-smoothing using pre-training with regression-based pre-training and adversarial fine-tuning. It is not clear from the paper why training a separate generative model is better than applying fine-tuning. **Secondly**, the authors claim to provide a theoretical argument based on equation (3) in the paper that links the proposed method to the optimal transport. However, the presented argument lacks proper rigour. The conclusion "... *the generative model can mainly focus on correcting the over-smoothed regions of the regression model output*" is substantiated only with an intuitive explanation and lacks a formal proof. **Thirdly**, the provided analysis assumes that the SE method is based on adversarial training, which excludes a large body of work [4, 5, 6, 7] that uses diffusion-based and bridge methods for SE. It is unclear how the conclusions from the paper can be applied to these methods.

3. The authors observe that the URGENT 2025 Challenge training dataset contains some recordings that degrade the performance of the SE model. Although potentially useful for challenge participants, this observation, in my opinion, constitutes only a marginal contribution. Moreover, to fully measure the effects of the degraded recordings, it would be beneficial to train various models -- both GAN-based and diffusion-based -- on the original and cleaned data. That would show that the impact of the degraded data is significant; otherwise, the loss in quality might be attributed to architectural inefficiencies and training setup

**Questions:**

How can the analysis of the trade-off between the generative and regression-based paradigms be generalised to other types of SE methods, such as bridge models or diffusion-based models?

#### **References:**

[1] Andreev et al., "HiFi++: a unified framework for bandwidth extension and speech enhancement".

[2] Babaev et al., "FINALLY: fast and universal speech enhancement with studio-like quality".

[3] Su et al., "HiFi-GAN-2: studio-quality speech enhancement via generative".

[4] Lemercier et al., "StoRM: a diffusion-based stochastic regeneration model for speech enhancement and dereverberation".

[5] Scheibler et al.,  "Universal score-based speech enhancement with high content preservation".

[6] Jukíc et al., "Schrödinger bridge for generative speech enhancement".

[7] Wang et al., "Diffusion-based Speech Enhancement with Schrödinger Bridge and Symmetric Noise Schedule".

---

> ### Author Response · Authors · 2025-11-18
> **Response to the Weakness 1**
>
> _weakness: The paper claims that aligning target and input audio recordings improves the overall performance of the model. This is not a new observation; e.g. [2] also aligns target and input audios for training. Moreover, this observation is quite obvious, since any impulse response used for augmentation introduces a time shift that is known and can be manually corrected relatively simply. Therefore, the observation that using yields better results is not a novelty._
>
> => Since we could not find a related description in “FINALLY” [2], could you please **specify where** the authors mentioned that they also applied the aligned target for speech dereverb? We would be happy to include and discuss it in our revised version.
>
> It is worth noting that **many recent studies** on speech dereverberation [r1-r4] ([r3-r4] are observed from their GitHub code)—including the URGENT Challenges [r5]—still adopt early-reflected speech as learning targets. One of the main objectives of our paper is to explore what constitutes an optimal target for speech dereverberation and to demonstrate that time-shifted anechoic clean speech can be a more suitable choice. To the best of our knowledge, there is **no other related work** to discuss and **compare these two targets** (early-reflected vs time-shifted anechoic) and we show that early reflections still **harm both speech quality and downstream ASR performance.**
>
> **Since many speech dereverberation studies still use early-reflected speech as learning targets, we aim to highlight to the community that time-shifted anechoic clean speech may serve as a more suitable learning target.**
>
> Audio examples of the “Early-reflected” and “Shifted anechoic” targets are provided in the Supplementary Material with more examples on our **demo page**: https://anonymous.4open.science/w/USE-5232/, where we believe reviewers can clearly evaluate the quality differences.
>
>
> [r1] Rao, Nagashree KS, et al. "Low-Complexity Neural Speech Dereverberation With Adaptive Target Control."  IEEE ICASSP, 2025.
>
> [r2] Luo, Xiaoxue, et al. "On phase recovery and preserving early reflections for deep-learning speech dereverberation." The Journal of the Acoustical Society of America 155.1 (2024): 436-451, 2024.
>
> [r3] Wang, Jiahe, et al. "MeanSE: Efficient Generative Speech Enhancement with Mean Flows." arXiv preprint arXiv:2509.21214 (2025).
>
> [r4] Nakata, Wataru, et al. "Sidon: Fast and Robust Open-Source Multilingual Speech Restoration for Large-scale Dataset Cleansing." arXiv preprint arXiv:2509.17052 (2025).
>
> [r5] Kohei Saijo, et al. “Interspeech 2025 urgent speech enhancement challenge.” InProc.Interspeech,2025.

---

> > ### Comment · Reviewer_CvYq · 2025-11-26
> >
> > We would like to thank the authors for pointing out that many modern speech models do not align the input and target audio, as well as stating the resulting degradation in performance.
> >
> > Although we agree that using shifting targets increases the overall quality of the trained model, and some works do not include this data processing in their data pipeline, we believe that such a contribution is marginal. Although extremely valuable, in our opinion, it does not hold as a standalone contribution of the paper, as this trick is quite straightforward.
> >
> > For these reasons, we believe that the novelty claim related to the input and target audio alignment is marginal.

---

> > > ### Author Response · Authors · 2025-12-03
> > > **Response to the Weakness 1**
> > >
> > > We respectfully disagree with this comment and believe there may still be a misunderstanding regarding the contribution of this work. Our main point is not that modern speech models fail to **align** input and target signals—**most already do**. Instead, we highlight that many recent methods still use targets **convolved with early-reflection of RIRs** (e.g., 50 ms), which inherently includes reverberant components. Our results show that using only the direct-path component as the target leads to improved speech quality and better downstream ASR performance. Audio examples of the “Early-reflected” and “Shifted anechoic” targets are provided in on our demo page.

---

> ### Author Response · Authors · 2025-11-18
> **Response to the Weakness 2**
>
> _weakness 2-1: ...Other GAN-based methods deal with over-smoothing using pre-training with regression-based pre-training and adversarial fine-tuning. It is not clear from the paper why training a separate generative model is better than applying fine-tuning._
>
> => Thank you for pointing this out! Yes, the over-smoothing problem can be easily addressed with a generative model, but it comes at a **cost**. According to the **perception–distortion tradeoff theory** [1], improving perceptual quality (reduce over-smoothing) often comes at the cost of reduced fidelity (e.g., degrade in intrusive SE metrics). The key point is how should we obtain a better **“deal”** with less fidelity degradation. A recent theory [2] pointed out that: a better tradeoff—minimizing MSE while preserving perfect perception—can be achieved by **optimally transporting the posterior mean prediction (i.e., the MMSE estimate) to the true data distribution.** Following this theory, we first train a regression model to obtain the posterior mean prediction, then fix its weights and apply a generative model to optimally transport the distribution.
>
> In the new Fig. 7 of our revised paper, it can be clearly observed that throughout the **entire learning process**, the two-stage framework consistently achieves **lower magnitude, phase and time losses, as well as higher PESQ scores,** compared with pre-training with regression followed by adversarial fine-tuning.
>
> [1] Blau, Yochai, and Tomer Michaeli. "The perception-distortion tradeoff." CVPR. 2018.
>
> [2] Freirich, Dror, Tomer Michaeli, and Ron Meir. "A theory of the distortion-perception tradeoff in wasserstein space." Neurips, 2021.
>
> [3] Gulrajani, Ishaan, et al. "Improved training of wasserstein gans." Neurips, 2017.
>
> ===============================================================================================
>
> _weakness 2-2: Secondly, the authors claim to provide a theoretical argument based on equation (3) in the paper that links the proposed method to the optimal transport. However, the presented argument lacks proper rigour..._
>
> => We apologize for the unclear wording in our paper. We have revised the abstract and main text to avoid misunderstanding. What we intended to emphasize is that the proposed **two-stage** framework follows a recent finding [1] suggesting that a better tradeoff between perception and distortion can be achieved by **optimally transporting the posterior mean prediction (i.e., the MMSE estimate) to the true data distribution.** Therefore, we first train a regression model to obtain the posterior mean prediction, then fix its weights and apply a generative model to optimally transport the distribution. Since Wasserstein GANs are related to optimal transport [2] and offer efficient inference, we selected GANs for the generative stage. Other models, such as **flow matching**, can also approximate the optimal transport process and have been applied to image restoration under the same theoretical framework [3].
>
> In addition to the above theory, from a **model-structure perspective**, Equation (3) is intended to illustrate why, in this two-stage framework, the GAN can focus on correcting **over-smoothed regions** while **leaving other parts unchanged**—thereby preserving fidelity—provided that the discriminator is Lipschitz continuous.
>
> [1] Freirich, Dror, Tomer Michaeli, and Ron Meir. "A theory of the distortion-perception tradeoff in wasserstein space." Neurips, 2021.
>
> [2] Gulrajani, Ishaan, et al. "Improved training of wasserstein gans." Neurips, 2017.
>
> [3] Ohayon, Guy, Tomer Michaeli, and Michael Elad. "Posterior-mean rectified flow: Towards minimum mse photo-realistic image restoration." ICLR (2025).
>
> ==============================================================================================
>
> _weakness 2-3: ..It is unclear how the conclusions from the paper can be applied to these methods._
>
> => As mentioned earlier, our two-stage framework is based on the theory [1] that “a better tradeoff between perception and distortion can be achieved by optimally transporting the posterior mean prediction to the true data distribution.” Since diffusion-based and bridge methods can also be viewed as approximations of optimal transport, they can likewise serve as the second stage of the two-stage framework. In Posterior-Mean Rectified Flow [2], for example, **flow matching** was employed as the second stage for image restoration.
>
> [1] Freirich, Dror, Tomer Michaeli, and Ron Meir. "A theory of the distortion-perception tradeoff in wasserstein space." Neurips, 2021.
>
> [2] Ohayon, Guy, Tomer Michaeli, and Michael Elad. "Posterior-mean rectified flow: Towards minimum mse photo-realistic image restoration." ICLR (2025).

---

> > ### Comment · Reviewer_CvYq · 2025-11-26
> >
> > We would like to thank the authors for the elaborate answer to the raised point. We appreciate the explanation of the proposed solution, which is based on the idea of decoupling learning the posterior mean prediction, which can be successfully learnt with regression, and a generative model, which increases the overall perceptual quality.
> >
> > Although we do believe that the perception-distortion trade-off is an important issue for generative models, we think that the paper does not fully provide an experimental backing of the proposed remedy. To make the claim made in the paper stronger, it would be recommended to compare the proposed model to the existing state-of-the-art speech enhancement pipeline. However, no such comparison is made in the paper.
> >
> > Although, paper shows the benefits of the proposed two-component over the standard pretraining methods, the comparison does not include a well-established baseline. To make the contribution stronger, we would suggest benchmarking the proposed method against several established SE algorithms and on a number of well-known datasets.
> >
> > When it comes to the applicability of the proposed solution to diffusion-based, flow-matching-based and bridge-based models, the authors once again offer mostly verbal comments that are not substantiated by the empirical evaluations, which reduces the credibility of the claims made.
> >
> > In its current presentation, we think, the claim lacks proper experimental backing, although it definitely deserves attention.

---

> > > ### Author Response · Authors · 2025-12-03
> > > **Response to the Weakness 2**
> > >
> > > We respectfully disagree with this comment as we suspect the reviewer may have **overlooked** some relevant experimental results reported in the paper. The paper **does** include comparisons with existing state-of-the-art speech enhancement systems. Specifically, Table 1 compares our approach with the **top three ranked systems** in the URGENT Challenge 2025. Table 2 further evaluates our method against other popular open-source USE models, such as **ClearerVoice-Studio** and **ResembleEnhance**. Additionally, Table 3 provides a comparison with Google’s **Miipher-2**. We hope this clarification helps address the reviewer’s concern. In addition, we believe that the comparison between the proposed two-component framework and the standard pre-training approach, as shown in **Fig. 7**, demonstrates that our method achieves higher fidelity.
> > >
> > > As noted in both the paper and our earlier response, the two-stage framework is **motivated** by a recent theoretical result [1], which states that a better distortion–perception trade-off can be achieved by optimally transporting the posterior mean prediction (i.e., the MMSE estimate) to the true data distribution. This concept is relatively **straightforward**, and we would like to clarify once again that we are **not the authors of this theory**. While demonstrating the effectiveness of this theory across different types of generative models (e.g., diffusion-based, flow-matching, or bridge models) is certainly interesting, we believe such an investigation falls **beyond the scope of this work**. We, as **'users'** of this theory, design our model framework based on it and demonstrate that it works effectively and outperforms other baselines.
> > >
> > > As previously mentioned, other generative models—such as flow matching—can also approximate the optimal transport process and have already been applied to **image restoration** within the same theoretical framework [2]. Therefore, we do not fully understand why the reviewer expects us to provide results using multiple generative model families.
> > >
> > > [1] Freirich, Dror, Tomer Michaeli, and Ron Meir. "A theory of the distortion-perception tradeoff in wasserstein space." Neurips, 2021.
> > >
> > > [2] Ohayon, Guy, Tomer Michaeli, and Michael Elad. "Posterior-mean rectified flow: Towards minimum mse photo-realistic image restoration." ICLR (2025).

---

> ### Author Response · Authors · 2025-11-18
> **Response to the Weakness 3 and Questions**
>
> _weakness 3: The authors observe that the URGENT 2025 Challenge training dataset contains some recordings that degrade the performance of the SE model. Although potentially useful for challenge participants, this observation, in my opinion, constitutes only a marginal contribution..._
>
> => Thank you for pointing this out. While we understand your concern, we respectfully disagree with this statement. Although the idea of examining training data quality may seem straightforward, we would like to emphasize its importance—particularly for **commonly used datasets that are often assumed to be “clean”** (e.g., Libri-TTS, DNS5, etc.). Several low-quality samples filtered out by VQScore are included in the Supplementary Material, where reviewers can observe the extent of the quality issues. Given the presence of such low-quality data, we believe it can **negatively impact** the performance of SE models, regardless of whether they are based on GANs or diffusion methods.
>
> =================================================================================================
>
> _Questions: How can the analysis of the trade-off between the generative and regression-based paradigms be generalised to other types of SE methods, such as bridge models or diffusion-based models?_
>
> => The analysis of the tradeoff between generative and regression models is not tied to any specific generative framework. If the reviewer is referring to Equation (3), it is merely a model-structure perspective illustrating why the proposed two-stage framework can preserve fidelity. The core idea, however, is based on the theory of **“optimal transport to data distribution from the posterior mean prediction can achieve a better trade-off”.**

---

> ### Comment · Reviewer_CvYq · 2025-11-26
>
> We would like to thank the authors for the detailed and elaborate responses they provided. Unfortunately, despite many valuable clarifications made in the responses, we do not see enough reasons for raising the score. Therefore, we are going to maintain the score as it is.

---

### Official Review · Reviewer_UWo4 · 2025-10-29

**Soundness:** 2
**Presentation:** 2
**Contribution:** 1
**Rating:** 2
**Confidence:** 5

**Summary:**

The paper investigates three approaches that aim at improving the training of universal speech enhancement (USE) systems. These three approaches are (a) enforcing higher quality training data by curating files with a high VQScore, (b) employing unreverberated training targets instead of such including early reflections, and (c) a two-stage training framework that combines discriminative and generative architectures. The authors show that a smaller, but higher quality dataset improves performance, and that non-reverberated targets as well as their two-stage approach raise scores on non-intrusive and task-dependent metrics.

**Strengths:**

The author’s train of thought is well described and easy to follow. The paper compares with both state of the art and open-source models on a publicly available test set. Furthermore, the authors did state their intent to publish code, which would ensure reproducibility of the presented results.

**Weaknesses:**

While the paper is at first sight well written and presents interesting results, there are multiple shortcomings. A major issue lies with the novelty and presentation of the discussed three “critical aspects”:

1.	Training targets: There are two issues with the authors’ claim here. Firstly, the assumption that the use of slightly reverberated targets is due to the difficulty of removing them is disputable, especially since the authors themselves contradict that argument by presenting models trained on (time-shifted) anechoic speech as superior in performance. The main reason for maintaining early reflections in the targets is that they are regarded as beneficial to speech intelligibility [Bradley 2003]. Secondly, the anechoic clean speech approach in Figure 3 (b) without consideration of the direct path delay is no sensible approach to begin with. The proposed solution with time-shifted targets (or vice-versa, removing direct path delay in the room impulse response) is not novel, but should rather be the standard for any reasonable training employing anechoic targets.
[Bradley, et al. “ On the importance of early reflections for speech in rooms,” 2003.]

2.	Model architecture: The results in Table 1 are unconvincing. Employing the GAN correction degrades more metrics than it improves. If the authors had adopted the overall ranking from URGENT, as they did with the other metrics, these models would have fallen behind their regression-only counterparts. Furthermore, improving non-intrusive metrics, which cannot detect hallucinations, is not surprising for a generative model. Only the slight improvement in CAcc seems interesting. Regarding novelty, combining regression and generative stages is nothing new and has been done in much more sophisticated ways before (see UNIVERSE++).
[Scheibler et al., “Universal Score-based Speech Enhancement with High Content Preservation”, 2024.]

3.	Training data quality: No novelty at all. This is basically the same which was already investigated in more detail and with a more sophisticated data curation strategy in the paper by [Li 2025], which was even cited by the authors. Furthermore, why would the authors rely solely on the seemingly not entirely suitable VQScore when samples of the dataset could be out of domain for this evaluation as mentioned in Section 2.3 (e.g., expressive speech)?
[Li et al., “Less is More: Data Curation Matters in Scaling Speech Enhancement”, 2024.]

Furthermore, the presentation of results is questionable. Why would the authors present a table with 12 metrics, only to argue that 8 of them cannot be considered for comparison due to different training targets? The complete lack of comparable intrusive metrics greatly reduces the significance of the results. A subjective degradation category rating (DCR) listening test between the “Early reflected” and “Shifted anechoic” approaches could have helped.

**Questions:**

Minor remarks:
-	Section 2.2: missing article before bold phrases (twice)
-	Fonts in figures are often too small
-	Some figures are just screenshots; labels contain compression artifacts
-	Figure 1: output -> Output
-	Table 3 would be more conclusive if the output of the models regression stage would also be reported
-	Consistently missing capitalizations in references (e.g., line 513: Ecapa-tdnn – ECAPA-TDNN, line 518: Icassp -> ICASSP, line 636: perceptual –> Perceptual)
-	Inconsistent formatting of references, e.g., line 603 Rix [Rix et al.] vs line 673 [Zhao et al.] – both are ICASSP papers
-	Figures 6 and 7 are missing axis descriptions (time, frequency range)

---

> ### Author Response · Authors · 2025-11-18
> **Response to the Weakness 1**
>
> _Weakness: Training targets: There are two issues with the authors’ claim here. Firstly, the assumption that the use of slightly reverberated targets is due to the difficulty of removing them is disputable, especially since the authors themselves contradict that argument by presenting models trained on (time-shifted) anechoic speech as superior in performance...._
>
> => We apologize for the unclear wording in our paper. In fact, the statement that “the use of slightly reverberated targets is due to the difficulty of removing them” originates from **previous** studies [1–3], not from us. Specifically, in section 3 of [1], the authors mentioned **“Early reflections are much harder to remove, but have a lesser impact on quality than late reverberation; The difficulty of solving the problem leads to excessive artifacts in the enhanced speech.”** And in section I of [2], the authors claim that **“… As a result, they normally have a large prediction error, which may cause speech distortion. Since early reflections do not cause speech quality degradation, they are often preserved and only late reverberation are removed, such as in WPE [10, 11] and the spectral subtraction methods”.** The results using anechoic clean speech without considering the direct path delay in Figure 3(b) were included to **verify** the observations made in those prior works [1–3].
>
> It is worth noting that **many recent** studies on speech dereverberation [4-7] ([6-7] are observed from their GitHub code)—including the URGENT Challenges [8]—still adopt early-reflected speech as learning targets. One of the main objectives of our paper is to explore what constitutes an optimal target for speech dereverberation and to demonstrate that time-shifted anechoic clean speech can be a more suitable choice. To the best of our knowledge, there is **no other related work** to discuss and **compare these two targets** (early-reflected vs time-shifted anechoic) and we show that early reflections still **harm both speech quality and downstream ASR performance.**
>
> **Since many speech dereverberation studies still use early-reflected speech as learning targets, we aim to highlight to the community that time-shifted anechoic clean speech may serve as a more suitable learning target.**
>
> [1] Jean-MarcValin, et al. “To dereverb or not to dereverb? Perceptual studies on real-time  dereverberation targets. “2022.
>
> [2] Rui  Zhou, et al. “Speech dereverberation with a reverberation time shortening target.” ICASSP 2023.
>
> [3] YanZhao, et al. ”Monaural speech dereverberation using temporal convolutional networks  with self attention.” TASLP ,2020.
>
> [4] Rao, Nagashree KS, et al. "Low-Complexity Neural Speech Dereverberation With Adaptive Target Control."  IEEE ICASSP, 2025.
>
> [5] Luo, Xiaoxue, et al. "On phase recovery and preserving early reflections for deep-learning speech dereverberation." The Journal of the Acoustical Society of America 155.1 (2024): 436-451, 2024.
>
> [6] Wang, Jiahe, et al. "MeanSE: Efficient Generative Speech Enhancement with Mean Flows." arXiv preprint arXiv:2509.21214 (2025).
>
> [7] Nakata, Wataru, et al. "Sidon: Fast and Robust Open-Source Multilingual Speech Restoration for Large-scale Dataset Cleansing." arXiv preprint arXiv:2509.17052 (2025).
>
> [8] Kohei Saijo, et al. “Interspeech 2025 urgent speech enhancement challenge.” InProc.Interspeech,2025.

---

> > ### Comment · Reviewer_UWo4 · 2025-11-24
> > **Response to Response to Weakness 1**
> >
> > While it is true that the cited references mention the difficulty of removing early reflections, this is not the sole reason for keeping the early reflections. [1] states, directly above the cited passage, “Anechoic speech sounds highly unnatural”, which is a clear argument against the use of anechoic targets and also supported by their results. Moreover, the cited passage of [2] notes that early reflections do not cause speech quality degradation.
> > Also, the claim that no related work has compared the targets is wrong. Besides their proposed method, [2] compares direct path (=delayed anechoic, cf. Fig. 1) and direct path plus early reflections, where the models with training targets including early reflections consistently outperform those with only direct path.
> > [1] Jean-MarcValin, et al. “To dereverb or not to dereverb? Perceptual studies on real-time dereverberation targets. “2022.
> > [2] Rui Zhou, et al. “Speech dereverberation with a reverberation time shortening target.” ICASSP 2023.

---

> > > ### Author Response · Authors · 2025-12-03
> > > **Response to Weakness 1**
> > >
> > > Thank you for raising this point. In fact, the sentence “Anechoic speech sounds highly unnatural” in [1] actually **supports our argument** rather than contradicts it. As defined in the paragraph preceding your cited sentence (“setting h_1 (t)=δ(t), where δ(t) is the Dirac function”), the term anechoic speech in [1] is defined specifically as **anechoic speech without time shift.** As noted in our previous response, the results using anechoic speech without direct-path delay (Fig. 3(b) in our paper) were included to validate and align with observations from prior works.
> > >
> > > In addition, in Section 3.4 of [1], the authors state that “In Fig. 3… Note that we did not include the early reflections component, i.e., the first 20 ms of RIR **since that is unchanged for all 4 cases.**” This means that early reflections are **always included** for all training target variants when studying late reverberation. In contrast, our paper makes a different argument: we propose removing the early-RIR component as well, and considering only time shift (δ(t-t0)) as the learning target.
> > >
> > > Yes, [2] does compare these two learning targets and arrives at a different conclusion. However, the scale and diversity of their training data are relatively limited—for example, only 7,861 clean utterances were used, and the training noise condition consisted **only** of air-conditioning noise. Such constraints may hinder the model’s ability to generalize well under broader testing scenarios.

---

> ### Author Response · Authors · 2025-11-18
> **Response to the Weakness 2**
>
> _Weakness 2-1: Model architecture: The results in Table 1 are unconvincing. Employing the GAN correction degrades more metrics than it improves._
>
> => Yes, according to the **perception–distortion tradeoff theory** [1], improving perceptual quality often comes at the cost of reduced fidelity (e.g., in intrusive SE metrics). However, the key point is how should we obtain a better **“deal”** with less fidelity degradation. A recent theory [2] mentioned that: a better tradeoff—minimizing MSE while preserving perfect perception—can be achieved by **optimally transporting the posterior mean prediction (i.e., the MMSE estimate) to the true data distribution.** Following this theory, we first train a regression model to obtain the posterior mean prediction, then fix its weights and apply a generative model to optimally transport the distribution. Since Wasserstein GANs are related to optimal transport [3] and offer efficient inference, we selected GANs for the generative stage.
>
> [1] Blau, Yochai, and Tomer Michaeli. "The perception-distortion tradeoff." CVPR. 2018.
>
> [2] Freirich, Dror, Tomer Michaeli, and Ron Meir. "A theory of the distortion-perception tradeoff in wasserstein space." Neurips, 2021.
>
> [3] Gulrajani, Ishaan, et al. "Improved training of wasserstein gans." Neurips, 2017.
>
> =================================================================================================
>
> _Weakness 2-2: If the authors had adopted the overall ranking from URGENT, as they did with the other metrics, these models would have fallen behind their regression-only counterparts._
>
> => Regarding the **overall ranking**, it must be evaluated in comparison with other teams across different **metrics** and **categories** (see the definition in section 3.5 of [1]) and therefore may not necessarily fall behind the regression-only counterparts (thanks to the limited fidelity degradation using the two-stage framework).
>
> [1] Kohei Saijo, et al. “Interspeech 2025 urgent speech enhancement challenge.” InProc.Interspeech,2025.
>
> =================================================================================================
>
> _Weakness 2-3: Furthermore, improving non-intrusive metrics, which cannot detect hallucinations, is not surprising for a generative model. Only the slight improvement in CAcc seems interesting._
>
> => As mentioned in the paper, the improvements in speaker similarity (SpkSim) and character accuracy (CAcc) implicitly confirm that the quality enhancements **do not come at the expense of hallucinations.**

---

> > ### Comment · Reviewer_UWo4 · 2025-11-24
> > **Response to Response to Weakness 2**
> >
> > 2-1: The motivation behind the proposed approach does not make the results more convincing. If the majority of reported metrics degrade and you are convinced to still obtain a better “deal”, a subjective CCR listening test would be necessary to prove the benefit for actual human subjects. The provided audio examples also do not include any comparison between the different approaches.
> > 2-2: So why do you not calculate the overall score to prove your point? All metrics and categories needed have already been reported.
> > 2-3: Improved speaker similarity does not necessarily mean no hallucinations. The slight increase in CAcc (less than 1% relative), as acknowledged before, does suggest an improvement, but when considering the degradation in all intrusive metrics, is not convincing enough. At least reporting the standard deviation to verify significance would be necessary.

---

> > > ### Author Response · Authors · 2025-12-03
> > > **Response to Weakness 2**
> > >
> > > 2-1: Thank you for the suggestion. We have added audio examples on the demo page to include comparisons across the different approaches.
> > >
> > > 2-2: The overall ranking for the top systems has been shown in the table below (and you can find the whole table in the demo page). From the table, we can observe that Our two-satge GAN correction can indeed **outperform** Our one-stage Early reflected.
> > > | Team          | DNSMOS    | NISQA      | UTMOS     | PESQ      | ESTOI     | SDR        | MCD        | LSD        | SpeechBERTScore | LPS       | SpkSim    | CAcc (%)   | Overall ranking score |
> > > |---------------|-----------|------------|-----------|-----------|-----------|------------|------------|------------|------------------|-----------|-----------|------------|------------------------|
> > > | Bobbsun       | 3.01 (8)  | 3.41 (6)   | 2.4 (3)   | 2.95 (1)  | 0.86 (1)  | 14.33 (1)  | 3.01 (4)   | 2.83 (5)   | 0.91 (1)         | 0.86 (1)  | 0.85 (1)  | 88.92 (1)  | 2.516           |
> > > | ****Our Early reflected + GAN correction****| 3.04 (5)  | 3.53 (3)   | 2.3 (6)   | 2.78 (4)  | 0.84 (4)  | 12.25 (5)  | 2.97 (3)   | 2.75 (4)   | 0.9 (2)          | 0.84 (3)  | 0.85 (1)  | 88.13 (2)  | 3.166           |
> > > | rc            | 3.01 (8)  | 3.21 (9)   | 2.3 (6)   | 2.79 (3)  | 0.85 (2)  | 13.11 (2)  | 2.93 (2)   | 2.94 (8)   | 0.9 (2)          | 0.85 (2)  | 0.84 (3)  | 88.05 (3)  | 4.016           |
> > > | ****Our Early reflected****| 3.06 (4) | 3.23 (8)   | 2.26 (8)  | 2.81 (2)  | 0.85 (2)  | 12.28 (4)  | 2.87 (1)   | 2.66 (1)   | 0.9 (2)          | 0.84 (3)  | 0.82 (5)  | 87.62 (5)  | 4.041           |
> > > | Xiaobin       | 3.0 (10)  | 3.45 (4)   | 2.31 (5)  | 2.74 (5)  | 0.84 (4)  | 13.06 (3)  | 3.3 (6)    | 3.08 (11)  | 0.89 (5)         | 0.84 (3)  | 0.83 (4)  | 87.94 (4)  | 5.033           |
> > > | subatomicseer | 3.02 (6)  | 3.28 (7)   | 2.34 (4)  | 2.63 (7)  | 0.82 (6)  | 12.18 (6)  | 3.9 (12)   | 3.06 (10)  | 0.88 (7)         | 0.82 (6)  | 0.82 (5)  | 86.15 (7)  | 6.591           |
> > >
> > > 2-3:
> > > As defined in previous studies [1][2], **hallucination** in speech enhancement refers to cases where the **lexical/linguistic content** or **speaker characteristics** do not match the original signal. In our work, both the content integrity (as reflected by ASR results, with ~**4% relative CER improvement**) and speaker characteristics (from 0.82 to 0.85) are improved through our GAN-based correction. Therefore, we believe this already demonstrates the absence of hallucination issues that may occur in purely generative models.
> > >
> > > [1] Kohei Saijo, et al. “Interspeech 2025 urgent speech enhancement challenge.” InProc.Interspeech,2025.
> > >
> > > [2] Scheibler, Robin, et al. "Universal score-based speech enhancement with high content preservation." Interspeech, 2024.

---

> ### Author Response · Authors · 2025-11-18
> **Response to the Weakness 2 (part 2)**
>
> _Weakness 2-4: Regarding novelty, combining regression and generative stages is nothing new and has been done in much more sophisticated ways before (see UNIVERSE++)._
>
> => Yes, there are lots of different combining methods. Recent studies [2-4] (see the Model Architecture Part in the Introduction section of our paper) have **heuristically** explored combining regression and generative models for USE in various ways, but **without theoretical support**. Motivated by the recent theoretical finding [1], we adopt a two-stage framework based on this principle. Therefore, one of the contributions of this paper is providing a practical implementation of this theory, demonstrating that it preserves fidelity better than other combination methods, as shown in the Table below.
>
> | Method            | DNSMOS | NISQA | UTMOS | PESQ | ESTOI | SDR   | MCD  | LSD  | SBERT | LPS  | SpkSim | CAcc  |
> |-------------------|--------|-------|-------|------|--------|-------|------|------|--------|------|---------|--------|
> | Our two-stage correction   | **3.04**   | **3.53**  | 2.30  | **2.78** | **0.84**   | 12.25 | **2.97** | **2.75** | **0.90**   | **0.84** | **0.85**    | **88.13** |
> | TS-URGENet [r2]   | 3.00   | 3.45  | 2.31  | 2.74 | 0.84   | **13.06** | 3.30 | 3.08 | 0.89   | 0.84 | 0.83    | 87.94 |
> | FUSE [r3]         | 3.02   | 3.28  | **2.34**  | 2.63 | 0.82   | 12.18 | 3.90 | 3.06 | 0.88   | 0.82 | 0.82    | 86.15 |
> | Le [r4]           | 2.96   | 3.15  | 2.18  | 2.44 | 0.82   | 12.09 | 3.28 | 3.27 | 0.88   | 0.82 | 0.82    | 86.46 |
>
>
> The table compares our simple two-stage framework (Rows 1) with other multi-stage frameworks proposed by teams that also participated in the URGENT Challenge, aiming to combine regression and generative models. Both TS-URGENet [2] and FUSE [3] adopt **three-stage** frameworks for generating the final USE results. Specifically, TS-URGENet [2] employs filling, separation, and restoration modules, while FUSE [3] uses a fusion network to combine the outputs of regression and token-sampling–based generative models. Le et al. [4] further propose a **four-stage** strategy consisting of audio declipping, packet loss compensation, audio separation, and spectral inpainting modules.
>
> From the results, we can observe that although our two-stage framework is simple, it outperforms other methods in most metrics and better preserves signal fidelity (from columns PESQ to LSD), as suggested by the theory.
>
> [1] Freirich, Dror, Tomer Michaeli, and Ron Meir. "A theory of the distortion-perception tradeoff in wasserstein space." Neurips, 2021.
>
> [2] Rong, Xiaobin, et al. "TS-URGENet: A Three-stage Universal Robust and Generalizable Speech Enhancement Network." Interspeech. 2025.
>
> [3] Goswami, Nabarun, and Tatsuya Harada. "FUSE: Universal Speech Enhancement using Multi-Stage Fusion of Sparse Compression and Token Generation Models for the URGENT 2025 Challenge." Interspeech. 2025.
>
> [4] Le, Xiaohuai, et al. "Multistage Universal Speech Enhancement System for URGENT Challenge." Proc. Interspeech. 2025.

---

> > ### Comment · Reviewer_UWo4 · 2025-11-24
> > **Response to the Response to Weakness 2 (part 2)**
> >
> > Previous  studies combining regression and generative models have provided extensive theoretical support before, e.g., by [Lemercier, 2023]. Also, as other reviewers have pointed out, the theoretical motivation is not sufficiently substantiated in this paper. How the distortion-perception trade-off motivates the (rather simple) proposed approach is not properly explained.
> > Furthermore, the comparison in the provided table (as well as Table 1 in your paper, by the way) is not fair, as your model was trained with the adopted improved dataset while the others were not. Therefore, this does not prove any advantage of your method.
> > [Lemercier et al., “StoRM: A Diffusion-based Stochastic Regeneration Model for Speech Enhancement and Dereverberation”, 2023.]

---

> > > ### Author Response · Authors · 2025-12-03
> > > **Response to Weakness 2 (part 2)**
> > >
> > > Thank you for pointing this out. We revisited the StoRM paper carefully, but did not find theoretical justification for the way **regression and generative models are combined**. Specifically, Equations (1)–(10) introduce score-based diffusion models, Equations (11)–(15) describe the previously proposed stochastic refinement and its limitations, and Equations (16)–(17) present the proposed stochastic regeneration method. However, no formal theoretical support is provided for this **combination**.
> > >
> > > Although the reviewer requested a further explanation of our two-stage framework, it is motivated by a recent theoretical result [1], which states that a better distortion–perception trade-off can be achieved by **optimally transporting the posterior mean prediction (i.e., the MMSE estimate) to the true data distribution**. We believe this concept is relatively **straightforward** — the MMSE estimate can be obtained using a regression model, and the optimal transport process can be approximated using generative models such as Wasserstein GANs [2] or rectified flows [3]. Since the underlying theory **has been clearly presented in prior work** [1–3], and we have cited and discussed these studies in our paper, we believe the motivation and reasoning behind our two-stage framework are sufficiently explained.
> > >
> > > In the new Fig. 7 of our revised paper, it can be clearly observed that throughout the **entire learning process**, the two-stage framework consistently achieves **lower** magnitude, phase and time losses, as well as higher PESQ scores, compared with pre-training with regression followed by adversarial fine-tuning. I believe this is a fair comparison in terms of the **same training data** to show the advantages of our two-stage framework.
> > >
> > > [1] Freirich, Dror, Tomer Michaeli, and Ron Meir. "A theory of the distortion-perception tradeoff in wasserstein space." Neurips, 2021.
> > >
> > > [2] Arjovsky, Martin, Soumith Chintala, and Léon Bottou. "Wasserstein generative adversarial networks." International conference on machine learning. PMLR, 2017.
> > >
> > > [3] Ohayon, Guy, Tomer Michaeli, and Michael Elad. "Posterior-mean rectified flow: Towards minimum mse photo-realistic image restoration." ICLR (2025).

---

> ### Author Response · Authors · 2025-11-18
> **Response to the Weakness 3, 4 and Questions**
>
> _weakness 3: Training data quality: No novelty at all. This is basically the same which was already investigated in more detail and with a more sophisticated data curation strategy in the paper by [Li 2025], which was even cited by the authors..._
>
> => Thank you for pointing this out. While we understand your concern, we respectfully disagree with this statement. Although investigating training data quality is straightforward, to the best of our knowledge, **only** the **concurrent** work (according to the definition of “FAQ for Reviewers” of ICLR review guide: https://iclr.cc/Conferences/2026/ReviewerGuide) [Li 2025] (to appear in ASRU **this December**) has also explored this direction.
>
> Since the Urgent Challenge organizers had already filtered out some noisy samples based on the DNSMOS score [1], we further applied VQScore to remove the **remaining** low-quality speech. We chose VQScore because it has been shown to exhibit a high correlation with subjective quality scores [2] in previous Urgent Challenges, and it also offers efficient inference performance.
>
> We would like to draw the community’s attention to the importance of training data quality, **particularly for some commonly used datasets that are often assumed to be “clean” (e.g., Libri-TTS, DNS5, etc.).** Several low-quality samples filtered out by VQScore are provided in the Supplementary Material, where reviewers can observe the extent of the quality issues.
>
> [1] Kohei Saijo, et al. “Interspeech 2025 urgent speech enhancement challenge.” InProc.Interspeech,2025.
>
> [2] Wangyou Zhang, et al. “Lessons learned from the urgent2024 speech enhancement challenge. “
>
> ================================================================================================
>
> _weakness 4: Furthermore, the presentation of results is questionable. Why would the authors present a table with 12 metrics, only to argue that 8 of them cannot be considered for comparison due to different training targets?..._
>
> => We apologize for the unclear wording in our paper. What we intended to emphasize is that the shaded metrics in the last two rows of Table I are not directly comparable to the previous rows, as the **definition of “clean” speech differs.** These shaded metrics require a clean reference, and in the URGENT Challenge, the organizers used **early-reflected** speech as the “clean” reference for evaluation. Even if we were able to obtain the “true” clean speech from the organizers, comparisons such as PESQ(enhanced **anechoic** speech, clean **anechoic** speech) versus PESQ(enhanced **early-reflected** speech, clean **early-reflected** speech) would still not be directly comparable.
>
> Audio examples of the “Early-reflected” and “Shifted anechoic” targets are provided in the Supplementary Material with more examples on our **demo page**: https://anonymous.4open.science/w/USE-5232/, where we believe reviewers can clearly evaluate the quality differences.
>
> ================================================================================================
>
> _Questions: Minor remarks:..._
>
> => Thank you for the suggestion! We have corrected the typo, adjusted the figure font sizes, and updated the reference. Please see the revised paper with the changes marked in red.

---

> > ### Comment · Reviewer_UWo4 · 2025-11-24
> > **Response to the Response to the Weakness 3, 4 and Questions**
> >
> > 3: It is true that the authors are not expected to know and compare to any concurrent/unpublished paper, as would be the case for [Li2025]. Regardless of the matter of novelty, as previously mentioned (and not addressed), there are still flaws in using solely VQScore due to bad ratings for out-of-domain data.
> > [Li et al., “Less is More: Data Curation Matters in Scaling Speech Enhancement”, 2025.]
> > 4: This point was directed towards the presentation of results. If the results are not comparable, why report them and confuse the reader? Furthermore, some metrics would be very much comparable if clean speech was available, e.g., LPS – simply by always comparing against clean speech in evaluation, even when using reverberated targets. Furthermore, the provided “early-reflected” targets in the supplementary material clearly contain more than just early reflections – there seem to be also noise and/or distortions.
> > Regarding minor remarks: Almost none of the remarks have been implemented sufficiently. E.g., figure font sizes are still too small, the references are still awfully inconsistent and not correctly capitalized, grammatical errors have not been corrected.

---

> > > ### Author Response · Authors · 2025-12-03
> > > **Response to the Weakness 3, 4 and Questions**
> > >
> > > 3.  As noted in our previous response, we **did not rely solely** on VQScore; we also considered the **DNSMOS score**, which is the primary metric used in the original URGENT Challenge. For this reason, the second metric **does not need to be perfectly accurate**—its role is simply to identify the **remaining** low-quality samples that DNSMOS does not filter out. From this perspective, the performance of VQScore is sufficient for our purposes.
> > >
> > > 4.  Although the results are not directly comparable in the table, we still include them because we believe this will make it easier for others to compare their results with ours if they use the same learning targets.
> > >
> > > Thank you for pointing this out. We have made the reference formatting consistent and added more audio comparisons to our demo page. Please see the updated paper and demo link.

---

> > > ### Author Response · Authors · 2025-12-04
> > > **Response to the Weakness 3, 4 and Questions (part 2)**
> > >
> > > We have obtained the **anechoic clean speech** for the non-blind set from the organizers and report the results using it as the **reference** for computing the evaluation scores. However, due to the **volume normalization** process applied during data simulation—specifically, the use of a scale factor that is not provided (see the referenced code https://github.com/urgent-challenge/urgent2025_challenge/blob/daf1730cc11bf450d05c2d9e1d8bb3afdd63c427/simulation/simulate_data_from_param.py#L500) —we are unable to accurately recover the original amplitude. For this reason, volume-sensitive metrics (i.e., SDR, MCD, LSD) are excluded from our comparison.
> > >
> > > From the results in the table below, we can observe that using shifted anechoic speech as the learning target **significantly outperforms** using early-reflected speech. We hope this addresses the reviewer’s concerns.
> > >
> > >
> > > | Method                     | PESQ | ESTOI | SBERT | LPS  | SpkSim |
> > > |---------------------------|------|-------|-------|------|--------|
> > > | Early reflected + GAN     | 2.40 | 0.69  | 0.88  | 0.83 | 0.83   |
> > > | Shifted anechoic + GAN    | **2.71** | **0.78**  | **0.89**  | **0.86** | **0.84**   |

---

### Official Review · Reviewer_43Aa · 2025-11-01

**Soundness:** 3
**Presentation:** 3
**Contribution:** 2
**Rating:** 2
**Confidence:** 4

**Summary:**

This paper considers the problem of universal speech enhancement, which is to restore the quality of diverse degraded speech while preserving fidelity. To address the problem that the use of early-reflected speech as the training target harms perceptual quality, the authors consider time-shifted anechoic clean speech as the target. To address the problem that regression models preserve fidelity but produce over-smoothed outputs while generative models improve perceptual quality but risk hallucination, the authors introduce a two-stage training approach that first trains a regression model and then freezes it to train a generative model. Provided experiment results demonstrated the effectiveness of the proposed method.

**Strengths:**

The proposed method of using time-shifted anechoic clean speech as training target and the two-stage training approach is interesting, and the provided experimental results demonstrated the effectiveness of the proposed method.

**Weaknesses:**

The novelty of this paper is somewhat limited. The contributions of this paper seems incremental in system design rather than proposing fundamentally new ideas. The use of time-shifted anechoic clean speech as the training target is an incremental modification of existing practices. The two-stage training approach is not new, similar approaches have been explored in both speech and image enhancement, e.g., prior work such as [1] also consider such a two-stage training approach for speech enhancement.

While the authors claim to provide a theoretical analysis of the two-stage framework (in Abstract), this is not substantiated. Section 2.2 mostly summarizes conclusion from prior work rather than presenting any new theoretical results. In particular, eq. (3) is quite trivial and does not offer any deeper understanding of why or how the two-stage approach works. Therefore, it is difficult to consider this as a theoretical contribution.

[1] Huang J, Yan Z, et al. A Two-Stage Training Framework for Joint Speech Compression and Enhancement. arXiv preprint arXiv:2309.04132, 2023.

**Questions:**

See the above Weaknesses.

---

> ### Author Response · Authors · 2025-11-18
> **Response to the Weakness 1**
>
> _Weaknesses: The contributions of this paper seems incremental in system design rather than proposing fundamentally new ideas. The use of time-shifted anechoic clean speech as the training target is an incremental modification of existing practices._
>
> => It is worth noting that **many recent studies** on speech dereverberation [1-4] ([3-4] are observed from their GitHub code)—including the URGENT Challenges [5]—still adopt early-reflected speech as learning targets. One of the main objectives of our paper is to explore what constitutes an optimal target for speech dereverberation and to demonstrate that time-shifted anechoic clean speech can be a more suitable choice. To the best of our knowledge, there is **no other related work to discuss and compare these two targets (early-reflected vs time-shifted anechoic) and we show that early reflections still harm both speech quality and downstream ASR performance.**
>
> **Since many speech dereverberation studies still use early-reflected speech as learning targets, we aim to highlight to the community that time-shifted anechoic clean speech may serve as a more suitable learning target.**
>
> Audio examples of the “Early-reflected” and “Shifted anechoic” targets are provided in the Supplementary Material with more examples on our **demo page**: https://anonymous.4open.science/w/USE-5232/, where we believe reviewers can clearly evaluate the quality differences.
>
> [1] Rao, Nagashree KS, et al. "Low-Complexity Neural Speech Dereverberation With Adaptive Target Control."  IEEE ICASSP, 2025.
>
> [2] Luo, Xiaoxue, et al. "On phase recovery and preserving early reflections for deep-learning speech dereverberation." The Journal of the Acoustical Society of America 155.1 (2024): 436-451, 2024.
>
> [3] Wang, Jiahe, et al. "MeanSE: Efficient Generative Speech Enhancement with Mean Flows." arXiv preprint arXiv:2509.21214 (2025).
>
> [4] Nakata, Wataru, et al. "Sidon: Fast and Robust Open-Source Multilingual Speech Restoration for Large-scale Dataset Cleansing." arXiv preprint arXiv:2509.17052 (2025).
>
> [5] Kohei Saijo, et al. “Interspeech 2025 urgent speech enhancement challenge.” InProc.Interspeech,2025.

---

> ### Author Response · Authors · 2025-11-18
> **Response to the Weakness 2**
>
> _Weakness: The two-stage training approach is not new, similar approaches have been explored in both speech and image enhancement, e.g., prior work such as [1] also consider such a two-stage training approach for speech enhancement._
>
>
> => Thank you for pointing this out! We have included [1] in our revised paper. Please see the updated paper with the revisions marked in red. Recent studies [r2-r4] (see the Model Architecture Part in the Introduction section of our paper) have **heuristically** explored combining regression and generative models for USE in various ways, but without theoretical support. Motivated by the recent theoretical finding [r1] that **“optimally transporting the posterior mean prediction to the true data distribution can achieve a better tradeoff between quality and fidelity,”** we adopt a two-stage framework based on this principle. Therefore, one of the contributions of this paper is providing a practical implementation of this theory, demonstrating that it **preserves fidelity better** than other combination methods (as shown in the Table below).
>
> | Method            | DNSMOS | NISQA | UTMOS | PESQ | ESTOI | SDR   | MCD  | LSD  | SBERT | LPS  | SpkSim | CAcc  |
> |-------------------|--------|-------|-------|------|--------|-------|------|------|--------|------|---------|--------|
> | Our two-stage correction   | **3.04**   | **3.53**  | 2.30  | **2.78** | **0.84**   | 12.25 | **2.97** | **2.75** | **0.90**   | **0.84** | **0.85**    | **88.13** |
> | TS-URGENet [r2]   | 3.00   | 3.45  | 2.31  | 2.74 | 0.84   | **13.06** | 3.30 | 3.08 | 0.89   | 0.84 | 0.83    | 87.94 |
> | FUSE [r3]         | 3.02   | 3.28  | **2.34**  | 2.63 | 0.82   | 12.18 | 3.90 | 3.06 | 0.88   | 0.82 | 0.82    | 86.15 |
> | Le [r4]           | 2.96   | 3.15  | 2.18  | 2.44 | 0.82   | 12.09 | 3.28 | 3.27 | 0.88   | 0.82 | 0.82    | 86.46 |
>
> The table compares our simple two-stage framework (Rows 1) with other multi-stage frameworks proposed by teams that also participated in the URGENT Challenge, aiming to combine regression and generative models. Both TS-URGENet [r2] and FUSE [r3] adopt **three-stage** frameworks for generating the final USE results. Specifically, TS-URGENet [r2] employs filling, separation, and restoration modules, while FUSE [r3] uses a fusion network to combine the outputs of regression and token-sampling–based generative models. Le et al. [r4] further propose a **four-stage** strategy consisting of audio declipping, packet loss compensation, audio separation, and spectral inpainting modules.
>
> From the results, we can observe that although our two-stage framework is simple, it outperforms other methods in most metrics and better preserves signal fidelity (from columns PESQ to LSD), as suggested by the theory.
>
> [r1] Freirich, Dror, Tomer Michaeli, and Ron Meir. "A theory of the distortion-perception tradeoff in wasserstein space." Neurips, 2021.
>
> [r2] Rong, Xiaobin, et al. "TS-URGENet: A Three-stage Universal Robust and Generalizable Speech Enhancement Network." Interspeech. 2025.
>
> [r3] Goswami, Nabarun, and Tatsuya Harada. "FUSE: Universal Speech Enhancement using Multi-Stage Fusion of Sparse Compression and Token Generation Models for the URGENT 2025 Challenge." Interspeech. 2025.
>
> [r4] Le, Xiaohuai, et al. "Multistage Universal Speech Enhancement System for URGENT Challenge." Proc. Interspeech. 2025.

---

> ### Author Response · Authors · 2025-11-18
> **Response to the Weakness 3**
>
> _Weaknesses: While the authors claim to provide a theoretical analysis of the two-stage framework (in Abstract), this is not substantiated. Section 2.2 mostly summarizes conclusion from prior work rather than presenting any new theoretical results. In particular, eq. (3) is quite trivial and does not offer any deeper understanding of why or how the two-stage approach works. Therefore, it is difficult to consider this as a theoretical contribution._
>
> => We apologize for the unclear wording in our paper. We have revised the abstract from “We provide theoretical analysis and introduce a two-stage framework that effectively combines the strengths of both approaches” to **“We introduce a two-stage framework that effectively combines the strengths of both approaches, inspired by a recent theoretical finding”**. What we intended to emphasize is that the proposed **two-stage** framework follows a recent finding : “a better tradeoff between quality and fidelity can be achieved by optimally transporting the posterior mean prediction to the true data distribution.” Therefore, we first train a regression model to obtain the posterior mean prediction, then fix its weights and apply a generative model to optimally transport the distribution. Since Wasserstein GANs are related to optimal transport [r5] and offer efficient inference, we selected GANs for the generative stage.
>
> In addition to the above theory, from a **model-structure perspective**, Equation (3) is intended to illustrate why, in this two-stage framework, GAN can focus on correcting **over-smoothed regions** while **leaving other parts unchanged**—thereby preserving fidelity—provided that the discriminator is Lipschitz continuous.
>
> [r5] Gulrajani, Ishaan, et al. "Improved training of wasserstein gans." Neurips, 2017.

---

### Author Response · Authors · 2025-11-18
**Summary of the Contributions of This Paper**

We thank all reviewers for their valuable comments and have uploaded the revised paper with more audio examples in our demo page.

In summary, this paper aims to encourage the related community to **rethink** some conventional practices that may not be optimal. By adopting **better learning targets** and **improving training data quality,** and by clarifying the roles of regression and generative models through theoretical support for achieving a **better distortion–perception tradeoff**, our simple yet effective model achieves **state-of-the-art performance** across several metrics in the URGENT Challenge benchmark.

1.	**Training target analysis**: We demonstrate that **time-shifted anechoic clean** speech is a more suitable training target for speech dereverberation than conventional **early-reflected speech.** While some prior works have used time-shifted anechoic speech as a target, to the best of our knowledge, **none** have explicitly discussed or compared these two target types. **Since many existing speech dereverberation studies still adopt early-reflected speech as the learning target, we aim to highlight the importance of this choice.**

2.	**Two-stage framework design:** Recent studies (see the Model Architecture Part in the Introduction section of our paper) have **heuristically** explored combining regression and generative models in various ways, but without theoretical support. Motivated by the recent theoretical finding that **“optimally transporting the posterior mean prediction to the true data distribution can achieve a better tradeoff between quality and fidelity,”** we adopt a simple two-stage framework based on this principle and show it can better **keep fidelity** than other combining methods.

3.	**Training data quality**: We want to draw the community’s attention to the importance of training data quality, particularly for widely used datasets that are often **assumed to be “clean”** (e.g., Libri-TTS, DNS5, etc.). As illustrated in Figure 8, models trained on such data struggle to remove artifacts like electrical microphone hiss, underscoring the need for higher-quality training data.

---

### Meta-Review · Area_Chair_sTtp · 2026-01-02

**Summary:**

The work is based on an existing idea, and therefore the novelty is limited. Experimental results not fully convincing (degradation in all intrusive metrics). Lack of substantiate theoretical justification for the proposed solution. In general the observations provided by the authors are not fully novel.

**Reviewer Concerns:**

The authors’ responses to key concerns are generally not fully convincing, and the main issues, regarding novelty, the lack of substantial theoretical justification for the proposed work, and insufficient experimental evaluation, remain outstanding.

**Reviewer Scores:**

43Aa may have not changed their score.

UWo4 may have not changed their score.

CvYq may have not changed their score.

---

### Decision · Program_Chairs · 2026-01-26

Reject